# Expressive power of tensor-network factorizations for probabilistic modeling

**Ivan Glasser[1,2]\*, Ryan Sweke[3], Nicola Pancotti[1,2], Jens Eisert[3,4], J. Ignacio Cirac[1,2]**
[1]Max-Planck-Institut für Quantenoptik, D-85748 Garching
[2]Munich Center for Quantum Science and Technology (MCQST), D-80799 München
[3]Dahlem Center for Complex Quantum Systems, Freie Universität Berlin, D-14195 Berlin
[4]Department of Mathematics and Computer Science, Freie Universität Berlin, D-14195 Berlin

## Abstract

Tensor-network techniques have recently proven useful in machine learning, both as a tool for the formulation of new learning algorithms and for enhancing the mathematical understanding of existing methods. Inspired by these developments, and the natural correspondence between tensor networks and probabilistic graphical models, we provide a rigorous analysis of the expressive power of various tensor-network factorizations of discrete multivariate probability distributions. These factorizations include non-negative tensor-trains/MPS, which are in correspondence with hidden Markov models, and Born machines, which are naturally related to the probabilistic interpretation of quantum circuits. When used to model probability distributions, they exhibit tractable likelihoods and admit efficient learning algorithms. Interestingly, we prove that there exist probability distributions for which there are unbounded separations between the resource requirements of some of these tensor-network factorizations. Of particular interest, using complex instead of real tensors can lead to an arbitrarily large reduction in the number of parameters of the network. Additionally, we introduce locally purified states (LPS), a new factorization inspired by techniques for the simulation of quantum systems, with provably better expressive power than all other representations considered. The ramifications of this result are explored through numerical experiments.

## 1 Introduction

Many problems in diverse areas of computer science and physics involve constructing efficient representations of high-dimensional functions. Neural networks are a particular example of such representations that have enjoyed great empirical success, and much effort has been dedicated to understanding their expressive power - i.e. the set of functions that they can efficiently represent. Analogously, tensor networks are a class of powerful representations of high-dimensional arrays (tensors), for which a variety of algorithms and methods have been developed. Examples of such tensor networks are tensor trains/matrix product states (MPS) [1, 2] or the hierarchical Tucker decomposition [3, 4], which have found application in data compression [5–7], the simulation of physical systems [8–10] and the design of machine learning algorithms [11–16]. In addition to their use in numerical algorithms, tensor networks enjoy a rich analytical understanding which has facilitated their use as a tool for obtaining rigorous results on the expressive power of deep learning models [17–22], and fundamental insights into the structure of quantum mechanical systems [23].

In the context of probabilistic modeling, tensor networks have been shown to be in natural correspondence with probabilistic graphical models [24–29], as well as with Sum-Product Networks and

Arithmetic Circuits [17, 30, 31]. Motivated by this correspondence, and with the goal of enhancing the toolbox for deriving analytical results on the properties of machine learning algorithms, we study the expressive power of various tensor-network models of discrete multivariate probability distributions. The models we consider, defined in Section 2, fall into two main categories:

- **Non-negative tensor networks**, which decompose a probability mass function as a network of non-negative tensors [32], as in a probabilistic graphical model [33].
- **Born machines (BM)**, which model a probability mass function as the absolute value squared of a real or complex function, which is itself represented as a network of real or complex tensors. While Born machines have been previously employed for probabilistic modeling [34–40], they have additional potential applications in the context of quantum machine learning [41–44], since they arise naturally from the probabilistic interpretation of quantum mechanics.

These models are considered precisely because they represent non-negative tensors by construction. In this work we focus on tensor networks which are based on tensor-trains/MPS and generalizations thereof, motivated by the fact that these have tractable likelihood, and thus efficient learning algorithms, while lending themselves to a rigorous theoretical analysis. In this setting non-negative tensor networks encompass hidden Markov models (HMM), while Born machines include models that arise from local quantum circuits of fixed depth. Our results also apply to tensor networks with a tree structure, and as such can be seen as a more general comparison of the difference between non-negative tensor networks and Born machines.

The main result of this work is a characterization of the expressive power of these tensor networks. Interestingly, we prove that there exist families of probability distributions for which there are unbounded separations between the resource requirements of some of these tensor-network factorizations. This allows us to show that neither HMM nor Born machines should be preferred to each other in general. Moreover, we prove that using complex instead of real tensors can sometimes lead to an arbitrarily large reduction in the number of parameters of the network.

Furthermore, we introduce a new tensor-network model of discrete multivariate probability distributions with provably better expressive power than the previously introduced models. This tensor network, which retains an efficient learning algorithm, is referred to as a locally purified state (LPS) due to its origin in the classical simulation of quantum systems [45–48]. We demonstrate through numerical experiments on both random probability distributions as well as realistic data sets that our theoretical findings are relevant in practice - i.e. that LPS should be considered over HMM and Born machines for probabilistic modeling.

This paper is structured as follows: The models we consider are introduced in Section 2. Their relation with HMM and quantum circuits is made explicit in Section 3. The main results on expressive power are presented in Section 4. Section 5 then introduces learning algorithms for these tensor networks, and the results of numerical experiments are provided in Section 6.

## 2 Tensor-network models of probability distributions

Consider a multivariate probability mass function $P(X_1, \ldots, X_N)$ over $N$ discrete random variables $\{X_i\}$ taking values in $\{1, \ldots, d\}$. This probability mass function is naturally represented as a multi-dimensional array, or tensor, with $N$ indices, each of which can take $d$ values. As such, we use the notation $P$ to refer simultaneously to both the probability mass function and the equivalent tensor representation. More specifically, for each configuration $X_1, \ldots, X_N$ the tensor element $P_{X_1,\ldots,X_N}$ stores the probability $P(X_1, \ldots, X_N)$. Note that as $P$ is a representation of a probability mass function, it is a tensor with non-negative entries summing to one.

Here we are interested in the case where $N$ is large. Since the number of elements of this tensor scales exponentially with $N$, it is quickly impossible to store. In cases where there is some structure to the variables, one may use a compact representation of $P$ which exploits this structure, such as Bayesian networks or Markov random fields defined on a graph. In the following we consider models, known as tensor networks, in which a tensor $T$ is factorized into the contraction of multiple smaller tensors. As long as $T$ is non-negative, one can model $P$ as $P = T/Z_T$, where $Z_T = \sum_{X_1,\ldots,X_N} T_{X_1,\ldots,X_N}$ is a normalization factor. For all tensor networks considered in this work, this normalization factor can be evaluated efficiently, as explained in Section 5.

In particular, we define the following tensor networks, in both algebraic and graphical notation. In the diagrams each box represents a tensor and lines emanating from these boxes represent tensor indices. Connecting two lines implies a contraction, which is a summation over the connected index.

1. **Tensor-train/matrix product state (MPS$_\mathbb{F}$):** A tensor $T$, with $N$ $d$-dimensional indices, admits an MPS$_\mathbb{F}$ representation of TT-rank$_\mathbb{F}$ $r$ when the entries of $T$ can be written as

$$T_{X_1,\ldots,X_N} \quad = \quad \sum_{\{\alpha_i=1\}}^{r} A_{1,X_1}^{\alpha_1} A_{2,X_2}^{\alpha_1,\alpha_2} \cdots A_{N-1,X_{N-1}}^{\alpha_{N-2},\alpha_{N-1}} A_{N,X_N}^{\alpha_{N-1}}, \qquad (1)$$

$$(2)$$

where $A_1$ and $A_N$ are $d \times r$ matrices, and $A_i$ are order-3 tensors of dimension $d \times r \times r$, with elements in $\mathbb{F} \in \{\mathbb{R}_{\geq 0}, \mathbb{R}, \mathbb{C}\}$. The indices $\alpha_i$ of these constituent tensors run from 1 to $r$ and are contracted (summed over) to construct $T$.

2. **Born machine (BM$_\mathbb{F}$):** A tensor $T$, with $N$ $d$-dimensional indices, admits a BM$_\mathbb{F}$ representation of Born-rank$_\mathbb{F}$ $r$ when the entries of $T$ can be written as

$$T_{X_1,\ldots,X_N} \quad = \quad \left| \sum_{\{\alpha_i=1\}}^{r} A_{1,X_1}^{\alpha_1} A_{2,X_2}^{\alpha_1,\alpha_2} \cdots A_{N-1,X_{N-1}}^{\alpha_{N-2},\alpha_{N-1}} A_{N,X_N}^{\alpha_{N-1}} \right|^2, \qquad (3)$$

$$(4)$$

with elements of the constituent tensors $A_i$ in $\mathbb{F} \in \{\mathbb{R}, \mathbb{C}\}$, i.e., when $T$ admits a representation as the absolute-value squared (element-wise) of an MPS$_\mathbb{F}$ of TT-rank$_\mathbb{F}$ $r$.

3. **Locally purified state (LPS$_\mathbb{F}$):** A tensor $T$, with $N$ $d$-dimensional indices, admits an LPS$_\mathbb{F}$ representation of puri-rank$_\mathbb{F}$ $r$ and purification dimension $\mu$ when the entries of $T$ can be written as

$$T_{X_1,\ldots,X_N} = \sum_{\{\alpha_i,\alpha_i'=1\}}^{r} \sum_{\{\beta_i=1\}}^{\mu} A_{1,X_1}^{\beta_1,\alpha_1} \overline{A_{1,X_1}^{\beta_1,\alpha_1'}} A_{2,X_2}^{\beta_2,\alpha_1,\alpha_2} \overline{A_{2,X_2}^{\beta_2,\alpha_1',\alpha_2'}} \cdots A_{N,X_N}^{\beta_N,\alpha_{N-1}} \overline{A_{N,X_N}^{\beta_N,\alpha_{N-1}'}},$$

$$(5)$$

$$(6)$$

where $A_1$ and $A_N$ are order-3 tensors of dimension $d \times \mu \times r$ and $A_i$ are order-4 tensors of dimension $d \times \mu \times r \times r$. The indices $\alpha_i$ run from 1 to $r$, the indices $\beta_i$ run from 1 to $\mu$, and both are contracted to construct $T$. Without loss of generality we can consider only $\mu \leq rd^2$.

Note that all the representations defined above yield non-negative tensors by construction, except for MPS$_{\mathbb{R}/\mathbb{C}}$. In this work, we consider only the subset of MPS$_{\mathbb{R}/\mathbb{C}}$ which represent non-negative tensors.

Given a non-negative tensor $T$ we define the TT-rank$_\mathbb{F}$ (Born-rank$_\mathbb{F}$) of $T$ as the minimal $r$ such that $T$ admits an MPS$_\mathbb{F}$ (BM$_\mathbb{F}$) representation of TT-rank$_\mathbb{F}$ (Born-rank$_\mathbb{F}$) $r$. We define the puri-rank$_\mathbb{F}$ of $T$

as the minimal $r$ such that $T$ admits an LPS$_\mathbb{F}$ representation of puri-rank$_\mathbb{F}$ $r$, for some purification dimension $\mu$. We note that if we consider tensors $T$ with 2 $d$-dimensional indices (i.e., matrices) then the TT-rank$_{\mathbb{R}_{\geq 0}}$ is the non-negative rank, i.e., the smallest $k$ such that $T$ can be written as $T = AB$ with $A$ being $d \times k$ and $B$ being $k \times d$ matrices with real non-negative entries. The TT-rank$_{\mathbb{R}/\mathbb{C}}$ is the conventional matrix rank, the Born-rank$_\mathbb{R}$ (Born-rank$_\mathbb{C}$) is the real (complex) Hadamard square-root rank, i.e., the minimal rank of a real (complex) entry-wise square root of $T$, and finally the puri-rank$_\mathbb{R}$ (puri-rank$_\mathbb{C}$) is the real (complex) positive semidefinite rank [49].

Table 1: Summary of notations for the different tensor-network representations and their ranks.

| Tensor representation | MPS$_{\mathbb{R}\geq 0}$ | MPS$_{\mathbb{R}/\mathbb{C}}$ | BM$_{\mathbb{R}/\mathbb{C}}$ | LPS$_{\mathbb{R}/\mathbb{C}}$ |
|---|---|---|---|---|
| Tensor rank | TT-rank$_{\mathbb{R}\geq 0}$ | TT-rank$_{\mathbb{R}/\mathbb{C}}$ | Born-rank$_{\mathbb{R}/\mathbb{C}}$ | puri-rank$_{\mathbb{R}/\mathbb{C}}$ |
| Matrix rank [49] | rank$_+$ | rank | rank$_{\mathbb{R}/\mathbb{C}\sqrt{}}$ | rank$_{\mathbb{R}/\mathbb{C},psd}$ |

For a given rank and a given tensor network, there is a set of non-negative tensors that can be exactly represented, and as the rank is increased, this set grows. In the limit of arbitrarily large rank, all tensor networks we consider can represent any non-negative tensor. This work is concerned with the relative expressive power of these different tensor-network representations, i.e. how do these representable sets compare for different tensor networks. This will be characterized in Section 4 in terms of the different ranks needed by different tensor networks to represent a non-negative tensor.

## 3 Relationship to hidden Markov models and quantum circuits

In order to provide context for the factorizations introduced in Section 2, we show here how they are related to other representations of probability distributions based on probabilistic graphical models and quantum circuits. In particular, we show that there is a mapping between hidden Markov models with constant number of hidden units per variable and MPS$_{\mathbb{R}\geq 0}$ with constant TT-rank$_{\mathbb{R}\geq 0}$, as well as between local quantum circuits of fixed depth and Born machines of constant Born-rank$_\mathbb{C}$. These relations imply that results on the expressive power of the former directly provide results on the expressive power of the latter.

### 3.1 Hidden Markov models are non-negative matrix product states

Consider a hidden Markov model (HMM) with observed variables $\{X_i\}$ taking values in $\{1, \ldots, d\}$ and hidden variables $\{H_i\}$ taking values in $\{1, \ldots, r\}$ (Fig. 1). The probability of the observed variables may be expressed as

$$P(X_1, \ldots, X_N) = \sum_{H_1, \ldots, H_N} P(X_1|H_1) \prod_{i=2}^{N} P(H_i|H_{i-1})P(X_i|H_i). \tag{7}$$

Notice that $P(H_i|H_{i-1})$ and $P(X_i|H_i)$ are matrices with non-negative elements, as depicted in the factor graph in the central diagram of Fig. 1. Now define the tensors $A_{1,l}^j = P(X_i = l|H_1 = j)$, and $A_{i,l}^{jk} = P(H_i = k|H_{i-1} = j)P(X_i = l|H_i = k)$. Then the MPS with TT-rank$_{\mathbb{R}_{\geq 0}} = r$ defined with tensors $A_i$ defines the same probability distribution on the observed variables as the HMM.

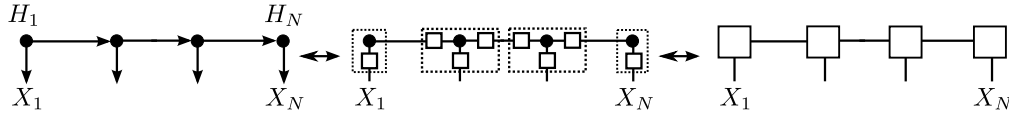

Figure 1: Mapping between a HMM and a non-negative MPS.

Conversely, given an MPS$_{\mathbb{R}\geq 0}$ with TT-rank$_{\mathbb{R}_{\geq 0}} = r$, there exists an HMM, with hidden variables of dimension $r' \leq \min(dr, r^2)$, defining the same probability mass function, as shown in the supplementary material. We note also that by using a different graph for the HMM, it is possible to construct an equivalent HMM with hidden variables of dimension $r$ [28, 29]. As such, any results on expressivity derived for MPS$_{\mathbb{R}\geq 0}$ hold also for HMM.

## 3.2 Quantum circuits are Born machines or locally purified states

An introductory presentation of the details of the connection between quantum circuits and Born machines is contained in the supplementary material. There, we show that local quantum circuits of fixed depth $D$ allow sampling from the probability mass function of $N$ discrete $d$-dimensional random variables $\{X_i\}$ which is given by the modulus squared of the amplitudes defined by the quantum circuit. For local quantum circuits of fixed depth $D$, these can be written as an MPS of TT-rank$_{\mathbb{C}} = d^{D+1}$. Therefore quantum circuits of fixed depth are in correspondence with Born machines of constant Born-rank$_{\mathbb{C}}$, and any results on the expressive power of Born machines hold also for local quantum circuits, when considered as probabilistic models.

Furthermore, quantum circuits that include alternating ancillary (or "hidden") and visible variables allow to sample from a probability distribution that can be expressed as a LPS. As such, this correspondence implies that any results on the expressive power of LPS hold also for local quantum circuits with alternating visible and hidden variables.

## 4 Expressive power of tensor-network representations

In this section we present various relationships between the expressive power of all representations, which constitute the primary results of this work. The proofs of the propositions in this section can be found in the supplementary material.

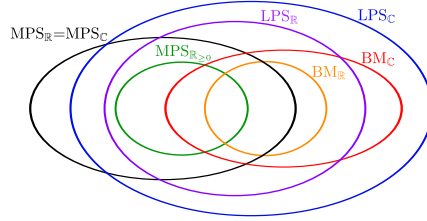

Figure 2: Representation of the sets of non-negative tensors that admit a given tensor-network factorization. In this figure we fix the different ranks of the different tensor networks to be equal.

For a given rank, there is a set of non-negative tensors that can be exactly represented by a given tensor network. These sets are represented in Fig. 2 for the case in which the ranks of the tensor networks are equal. When one set is included in another, it means that for every non-negative tensor, the rank of one of the tensor-network factorizations is always greater than or equal to the rank of the other factorization. The inclusion relationships between these sets can therefore be characterized in terms of inequalities between the ranks, as detailed in Proposition 1.

**Proposition 1.** *For all non-negative tensors TT-rank$_{\mathbb{R}_{\geq 0}} \geq$ TT-rank$_{\mathbb{R}}$, Born-rank$_{\mathbb{R}} \geq$ Born-rank$_{\mathbb{C}}$, Born-rank$_{\mathbb{R}} \geq$ puri-rank$_{\mathbb{R}}$, Born-rank$_{\mathbb{C}} \geq$ puri-rank$_{\mathbb{C}}$, puri-rank$_{\mathbb{R}} \geq$ puri-rank$_{\mathbb{C}}$, TT-rank$_{\mathbb{R}_{\geq 0}} \geq$ puri-rank$_{\mathbb{R}}$, TT-rank$_{\mathbb{R}} =$ TT-rank$_{\mathbb{C}}$.*

Next, as detailed in Proposition 2, and summarized in Table 2, we continue by showing that all the inequalities of Proposition 1 can in fact be strict, and that for all other pairs of representations there exist probability distributions showing that neither rank can always be lower than the other. This shows that neither of the two corresponding sets of tensors can be included in the other. The main new result is the introduction of a matrix with non-negative rank strictly smaller than its complex Hadamard square-root rank, i.e. TT-rank$_{\mathbb{R}_{\geq 0}} <$ Born-rank$_{\mathbb{C}}$.

**Proposition 2.** *The ranks of all introduced tensor-network representations satisfy the properties contained in Table 2. Specifically, denoting by $r_{row}$ ($r_{column}$) the rank appearing in the row (column), $<$ indicates that there exists a tensor satisfying $r_{row} < r_{column}$ and $<, >$ indicates that there exists both a tensor satisfying $r_{row} < r_{column}$ and another tensor satisfying $r_{column} > r_{row}$.*

We now answer the question: By how much do we need to increase the rank of a tensor network such that the set of tensors it can represent includes the set of tensors that can be represented by a different tensor network of a different rank? More specifically, consider a tensor that has rank $r$ according to one representation and rank $r'$ according to another. Can we bound the rank $r$ as a function of the rank $r'$ only? The results of Proposition 3, presented via Table 3, indicate that in many cases there is

| | TT-rank$_\mathbb{R}$ | TT-rank$_{\mathbb{R}_{\geq 0}}$ | Born-rank$_\mathbb{R}$ | Born-rank$_\mathbb{C}$ | puri-rank$_\mathbb{R}$ | puri-rank$_\mathbb{C}$ |
|---|---|---|---|---|---|---|
| TT-rank$_\mathbb{R}$ | = | < | <,> | <,> | <,> | <,> |
| TT-rank$_{\mathbb{R}_{\geq 0}}$ | > | = | <,> | <,> | > | > |
| Born-rank$_\mathbb{R}$ | <,> | <,> | = | > | > | > |
| Born-rank$_\mathbb{C}$ | <,> | <,> | < | = | <,> | > |
| puri-rank$_\mathbb{R}$ | <,> | < | < | <,> | = | > |
| puri-rank$_\mathbb{C}$ | <,> | < | < | < | < | = |

no such function - i.e. there exists a family of non-negative tensors, describing a family of probability distributions over $N$ binary variables, with the property that as $N$ goes to infinity $r'$ remains constant, while $r$ also goes to infinity.

**Proposition 3.** *The ranks of all introduced tensor-network representations satisfy the relationships without asterisk contained in Table 3. A function $g(x)$ denotes that for all non-negative tensors $r_{row} \leq g(r_{column})$. "No" indicates that there exists a family of probability distributions of increasing $N$ with $d = 2$ and $r_{column}$ constant, but such that $r_{row}$ goes to infinity, i.e. that no such function can exist.*

Table 3: Results of Proposition 3.

| | TT-rank$_\mathbb{R}$ | TT-rank$_{\mathbb{R}_{\geq 0}}$ | Born-rank$_\mathbb{R}$ | Born-rank$_\mathbb{C}$ | puri-rank$_\mathbb{R}$ | puri-rank$_\mathbb{C}$ |
|---|---|---|---|---|---|---|
| TT-rank$_\mathbb{R}$ | = | $\leq x$ | $\leq x^2$ | $\leq x^2$ | $\leq x^2$ | $\leq x^2$ |
| TT-rank$_{\mathbb{R}_{\geq 0}}$ | No | = | No | No | No | No |
| Born-rank$_\mathbb{R}$ | No | No | = | No | No | No |
| Born-rank$_\mathbb{C}$ | No | No* | $\leq x$ | = | No* | No* |
| puri-rank$_\mathbb{R}$ | No | $\leq x$ | $\leq x$ | $\leq 2x$ | = | $\leq 2x$ |
| puri-rank$_\mathbb{C}$ | No | $\leq x$ | $\leq x$ | $\leq x$ | $\leq x$ | = |

We conjecture that the relationships with an asterisk in Table 3 also hold. The existence of a family of matrices with constant non-negative rank but unbounded complex Hadamard square-root rank, together with the techniques introduced in the supplementary material, would provide a proof of these conjectured results. Proposition 3 indicates the existence of various families of non-negative tensors for which the rank of one representation remains constant, while the rank of another representation grows with the number of binary variables, however, the rate of this growth is not given. The following propositions provide details of the asymptotic growth of these ranks.

**Proposition 4** ([46])**.** *There exists a family of non-negative tensors over $2N$ binary variables and constant TT-rank$_\mathbb{R}$=3 that have puri-rank$_\mathbb{C} = \Omega(N)$, and hence also puri-rank$_\mathbb{C}$, Born-rank$_{\mathbb{R}/\mathbb{C}}$ and TT-rank$_{\mathbb{R}_{\geq 0}} \geq \Omega(N)$.*

**Proposition 5.** *There exists a family of non-negative tensors over $2N$ binary variables and constant TT-rank$_{\mathbb{R}_{\geq 0}}$=2 (and hence also puri-rank$_{\mathbb{R}/\mathbb{C}} = 2$) that have Born-rank$_\mathbb{R} \geq \pi(2^{N+1})$, where $\pi(x)$ is the number of prime numbers up to $x$, which asymptotically satisfies $\pi(x) \sim x/\log(x)$.*

**Proposition 6.** *There exists a family of non-negative tensors over $2N$ binary variables and constant Born-rank$_\mathbb{R}$=2 (and hence also constant Born-rank$_\mathbb{C}$ and puri-rank$_{\mathbb{R}/\mathbb{C}}$) that have TT-rank$_{\mathbb{R}_{\geq 0}} \geq N$.*

**Proposition 7.** *There exists a family of non-negative tensors over $2N$ binary variables and constant Born-rank$_\mathbb{C}$=2 that have Born-rank$_\mathbb{R} \geq N$.*

Some comments and observations which may aid in facilitating an intuitive understanding of these results are as follows: Cancellations between negative contributions allow an MPS$_\mathbb{R}$ to represent a non-negative tensor while having lower rank than an MPS$_{\mathbb{R}_{\geq 0}}$ (this separation can also be derived from the separation between Arithmetic Circuits and Monotone Arithmetic Circuits [50]). The separations between MPS$_{\mathbb{R}_{\geq 0}}$ and BM$_{\mathbb{R}/\mathbb{C}}$ are due to the difference of rank between probability distributions and their real or complex square roots. Finally, the difference between real and complex BM is due to the way in which real and imaginary elements are combined through the modulus squared, and this is illustrated well by the fact that real LPS of purification dimension 2 include complex BM.

As the techniques via which the results of Proposition 3 have been obtained are of interest, we provide a sketch of the proof for all "No" entries here . Assume that for a given pair of representations there exists a family of non-negative matrices with the property that the rank $r_{\text{column}}$ of one representation remains constant as a function of matrix dimension, while the rank $r_{\text{row}}$ of the other representation grows. Now, consider such a matrix $M$ of dimension $2^N \times 2^N$. The first step is to show that $M$ can be unfolded into a tensor network of constant rank $r_{\text{column}}$, for $2N$ binary variables, such that $M$ is a reshaping of the central bipartition of this tensor as

$$M = \quad = \quad = \qquad . \tag{8}$$

If the rank $r_{\text{row}}$ of matrix $M$ is large, the rank $r_{\text{row}}$ of the corresponding tensor-network representation of the unfolded tensor will also be large. While above unfolding requires a particular matrix dimension, it is in fact possible to write any $N \times N$ matrix $M$ as a submatrix of a $2^N \times 2^N$ matrix, to which the above unfolding strategy can then be used as a tool for leveraging matrix rank separations [51, 52, 49, 53] into tensor rank separations [54].

Finally, in order to discuss the significance of these results, note firstly that the TT-rank$_{\mathbb{R}}$ can be arbitrarily smaller than all other ranks, however, optimizing a real MPS to represent a probability distribution presents a problem since it is not clear how to impose positivity of the contracted tensor network [25, 48]. All other separations are relevant in practice since, as discussed in the following section, they apply to tensor networks that can be trained to represent probability distributions over many variables. Taken together, these results then show that LPS should be preferred over MPS$_{\mathbb{R}_{\geq 0}}$ or BM, since the puri-ranks will always be lower bounded compared to the other ranks. Additionally, complex BM should also be preferred to real BM as they can lead to an arbitrarily large reduction in the number of parameters of the tensor network. Note that because of the structure of the tensor networks we consider, these results also apply to more general tensor factorizations relying on a tree structure of the tensor network. How these results are affected if one considers approximate as opposed to exact representations remains an interesting open problem.

## 5 Learning algorithms

While the primary results of this work concern the expressive power of different tensor-network representations of probability distributions, these results are relevant in practice since MPS$_{\mathbb{R}_{\geq 0}}$, BM$_{\mathbb{R}/\mathbb{C}}$ and LPS$_{\mathbb{R}/\mathbb{C}}$ admit efficient learning algorithms, as shown in this section.

First, given samples $\{\mathbf{x_i} = (X_1^i, \ldots, X_N^i)\}$ from a discrete multivariate distribution, they can be trained to approximate this distribution through maximum likelihood estimation. Specifically, this can be done by minimizing the negative log-likelihood,

$$L = -\sum_i \log \frac{T_{\mathbf{x_i}}}{Z_T}, \text{ with derivatives } \partial_w L = -\sum_i \frac{\partial_w T_{\mathbf{x_i}}}{T_{\mathbf{x_i}}} - \frac{\partial_w Z_T}{Z_T}, \tag{9}$$

where $i$ indexes training samples and $T_{\mathbf{x_i}}$ is given by the contraction of one of the tensor-network models we have introduced. The negative log-likelihood can be minimized using a mini-batch gradient-descent algorithm. Note that when using complex tensors, the derivatives are replaced by Wirtinger derivatives with respect to the conjugated tensor elements. This algorithm requires the computation of $T_{\mathbf{x_i}}$ and $\partial_w T_{\mathbf{x_i}}$ for a training instance, as well as of the normalization $Z_T$ and its derivative $\partial_w Z_T$. We first focus on the computation of these quantities for LPS. Since BM are LPS of purification dimension $\mu = 1$, they can directly use the same algorithm [34]. For an LPS$_{\mathbb{C}}$ of puri-rank $r$, the normalization $Z_T$ can be computed by contracting the tensor network

$$Z_T = \qquad , \text{ with derivatives } \frac{\partial Z_T}{\partial \bar{A}_{i,m}^{j,k,l}} = \boxed{E_{i-1}} \boxed{A_i} \boxed{F_{i+1}} , \tag{10}$$

where the tensors $E_i$ and $F_i$ are intermediate tensors obtained by contracting the left part and right part of the tensor network corresponding to the norm. All these computations can be performed

in $\mathcal{O}(d\mu r^3 N)$ operations, and a similar contraction with fixed values for $X_i$ is used for computing $T_{\mathbf{x_i}}$ and its derivative at a training example. More details about this algorithm are included in the supplementary material, together with the algorithm we use for training $\mathrm{MPS}_{\mathbb{R}_{\geq 0}}$, which is a variation of the one given above for LPS. $\mathrm{MPS}_{\mathbb{R}_{\geq 0}}$ could also be trained using the expectation-maximization (EM) algorithm, but as BM and LPS use real or complex tensors, different algorithms are required. Note that in all these models not only the likelihood can be evaluated efficiently: marginals and correlation functions can be computed in a time linear in the number of variables, while exact samples from the distribution can also be generated efficiently [55, 34].

Instead of approximating a distribution from samples, it might also be useful to compress a probability mass function $P$ given in the form of a non-negative tensor. Since the original probability mass function has a number of parameters that is exponential in $N$, this is only possible for a small number of variables. It can be done by minimizing the Kullback–Leibler (KL) divergence $D(P||T/Z_T) = \sum_{X_1,\ldots,X_N} P_{X_1,\ldots,X_N} \log\left(\frac{P_{X_1,\ldots,X_N}}{T_{X_1,\ldots,X_N}/Z_T}\right)$, where $T$ is represented by a tensor-network model. The gradient of the KL-divergence can be obtained in the same way as the gradient of the log-likelihood and gradient-based optimization algorithms can then be used to solve this optimization problem. Note that for the case of matrices and $\mathrm{MPS}_{\mathbb{R}_{\geq 0}}$ more specific algorithms have been developed [56], and finding more efficient algorithms for factorizing a given tensor in the form of a BM or LPS represents an interesting problem that we leave for future work.

## 6 Numerical experiments

Using the algorithms discussed in Section 5 we numerically investigate the extent to which the separations found in Section 4 apply in both the setting of approximating a distribution from samples, and the setting of compressing given non-negative tensors. Code, data sets and choice of hyperparameters are available in the supplementary material and the provided repository [57].

### 6.1 Random tensor factorizations

We first generate random probability mass functions $P$ by generating a tensor with elements chosen uniformly in $[0, 1]$ and normalizing it. We then minimize the KL-divergence $D(P||T/Z_T)$, where $T$ is the tensor defined by an MPS, BM or LPS with given rank $r$. We choose LPS to have a purification dimension of 2. Details of the optimization are available in the supplementary material.

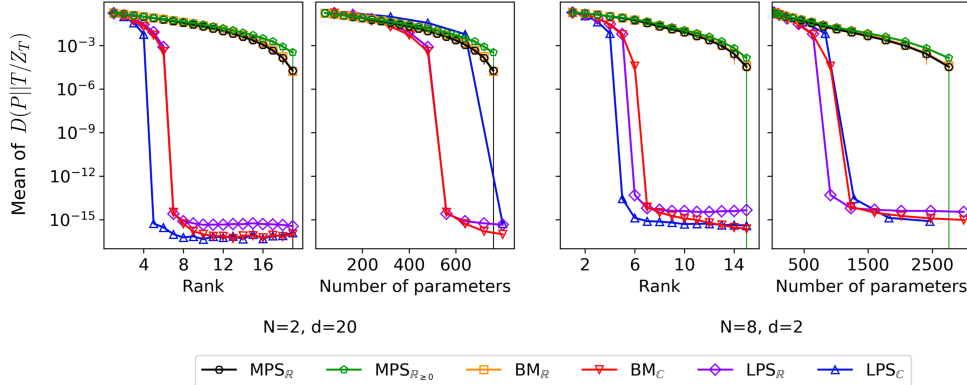

Figure 3: Mean of the minimum error of the approximation of 50 random tensors $P$ with tensor networks of fixed rank, as a function of the rank or the number of (real) parameters. Left: $20 \times 20$ matrix. Right: tensor over 8 binary variables. The errors bars represent one standard deviation, and are omitted below $10^{-12}$.

Results are presented in Fig. 3 for a $20 \times 20$ matrix and a tensor with 8 binary variables. They show that complex BM as well as real and complex LPS generically provide a better approximation to a tensor than an MPS or real BM, for fixed rank as well as for fixed number of real parameters.

## 6.2 Maximum likelihood estimation on realistic data sets

We now investigate how well the different tensor-network representations are able to learn from realistic data sets. We train $\text{MPS}_{\mathbb{R}_{\geq 0}}$, $\text{BM}_{\mathbb{R}}$, $\text{BM}_{\mathbb{C}}$, $\text{LPS}_{\mathbb{R}}$ and $\text{LPS}_{\mathbb{C}}$ (of purification dimension 2) using the algorithm of Section 5 on different data sets of categorical variables. Since we are interested in the expressive power of the different representations we use the complete data sets and no regularization. Additional results on generalization performance are included in the supplementary material.

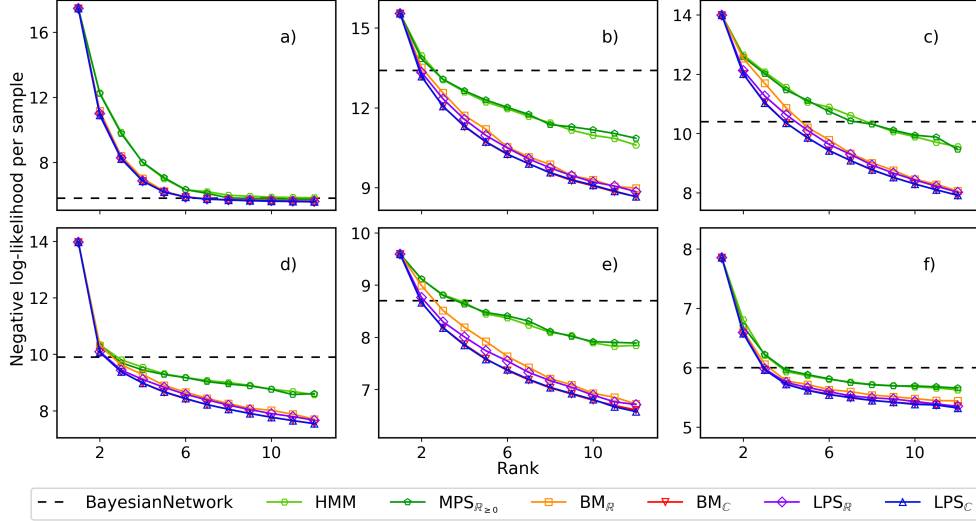

Figure 4: Maximum likelihood estimation with tensor networks, HMM and a Bayesian network without hidden units with graph learned from the data on different data sets: a) biofam data set of family life states from the Swiss Household Panel biographical survey [58]; data sets from the UCI Machine Learning Repository [59]: b) Lymphography [60], c) SPECT Heart, d) Congressional Voting Records, e) Primary Tumor [60], f) Solar Flare.

The results in Fig. 4 show the best negative log-likelihood per sample obtained for each tensor network of fixed rank. As a comparison we also include the best negative log-likelihood obtained from an HMM trained using the Baum-Welch algorithm [61, 62], as well as the best possible Bayesian network without hidden variables, where the network graph is learned from the data [62]. We observe that despite the different algorithm choice, the performance of HMM and $\text{MPS}_{\mathbb{R}_{\geq 0}}$ are similar, as we could expect from their relationship. On all data sets, BM and LPS lead to significant improvements for the same rank over $\text{MPS}_{\mathbb{R}_{\geq 0}}$.

## 7 Conclusion

We have characterized the expressive power of various tensor-network models of probability distributions, in the process enhancing the scope and applicability of the tensor-network toolbox within the broader context of learning algorithms. In particular, our analysis has concrete implications for model selection, suggesting that in generic settings LPS should be preferred over both hidden Markov models and Born machines. Furthermore, our results prove that unexpectedly the use of complex tensors over real tensors can lead to an unbounded expressive advantage in particular network architectures. Additionally, this work contributes to the growing body of rigorous results concerning the expressive power of learning models, which have been obtained via tensor-network techniques. A formal understanding of the expressive power of state-of-the-art learning models is often elusive; it is hoped that both the techniques and spirit of this work can be used to add momentum to this program. Finally, through the formal relationship of LPS and Born machines to quantum circuits, our work provides a concrete foundation for both the development and analysis of quantum machine learning algorithms for near-term quantum devices.

## Acknowledgments

We would like to thank Vedran Dunjko for his comments on the manuscript and João Gouveia for his suggestion of the proof of Lemma 9 in the supplementary material. I. G., N. P. and J. I. C. are supported by an ERC Advanced Grant QENOCOBA under the EU Horizon 2020 program (grant agreement 742102) and the German Research Foundation (DFG) under Germany's Excellence Strategy through Project No. EXC-2111 - 390814868 (MCQST). R. S. acknowledges the financial support of the Alexander von Humboldt foundation. N. P. acknowledges financial support from ExQM. J. E. acknowledges financial support by the German Research Foundation DFG (CRC 183 project B2, EI 519/7-1, CRC 1114, GRK 2433) and MATH+. This work has also received funding from the European Union's Horizon 2020 research and innovation programme under grant agreement No 817482 (PASQuanS).

## Footnotes

\*Corresponding author, `ivan.glasser@mpq.mpg.de`

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
