[Supplementary Material · SupplementaryMaterialNeurIPS2019.pdf]

# Supplementary material:

## Expressive power of tensor-network factorizations for probabilistic modeling

**Ivan Glasser[1,2]\*, Ryan Sweke[3], Nicola Pancotti[1,2], Jens Eisert[3,4], J. Ignacio Cirac[1,2]**
[1]Max-Planck-Institut für Quantenoptik, D-85748 Garching
[2]Munich Center for Quantum Science and Technology (MCQST), D-80799 München
[3]Dahlem Center for Complex Quantum Systems, Freie Universität Berlin, D-14195 Berlin
[4]Department of Mathematics and Computer Science, Freie Universität Berlin, D-14195 Berlin

## Contents

## 1 Relationship between tensor networks, hidden Markov models and quantum circuits

### 1.1 Non-negative MPS are HMM

Consider an MPS with non-negative tensors $A_i$ and TT-rank$_{\mathbb{R}_{\geq 0}} = r$. To express the corresponding probability distribution as a HMM, we split the tensors using an (exact) non-negative canonical polyadic decomposition such that $A_{i,l}^{jk} = \sum_{s=1}^{r'} B_i^{js} C_i^{ls} D_i^{sk}$, where $r' \leq \min(dr, r^2)$ (Fig. S1b). We can now set

$$P(X_i = l | H_i = s) = C_i^{ls} \tag{S1}$$

$$P(H_i = s | H_{i-1} = j) = \sum_u D_{i-1}^{ju} B_i^{us}, \tag{S2}$$

where the probabilities must be normalized properly, which can be done by first constructing the unnormalized factor graph and then normalizing the probabilities on every edge. We have then defined a HMM with hidden variables of dimension $r'$ that defines the same probability of the observed units as the one arising from the MPS. Note that using a different graph for the hidden Markov model we could also arrive at a dimension of hidden variables of $r$ [1, 2].

Figure S1: (a) Mapping of a hidden Markov chain to a MPS with non-negative tensor elements. (b) Mapping of a MPS with non-negative tensor elements to a hidden Markov chain.

## 1.2 Local quantum circuits are Born machines

In order to clarify the relationship between Born machines and local quantum circuits we provide here a concise introduction to the formalism of circuit based quantum computing. For a more thorough description, see ref. [3].

Consider the Hilbert space $\mathcal{H} = \mathbb{C}^d$, with ortho-normal basis $\{|X\rangle\}_{X=1}^d$, where Dirac notation $|X\rangle$ has been used to represent a vector $\vec{X} \in \mathcal{H}$. From a mathematical perspective, a $d$-dimensional *qudit* is a system whose state vector $|\psi\rangle$ can be described by a unit vector in $\mathcal{H}$. With respect to any fixed ortho-normal basis $\{|X\rangle\}_{X=1}^d$, a qudit is therefore specified by $d$ complex amplitudes $\{\psi(X)\}_{X=1}^d$ - i.e., $|\psi\rangle = \sum_{X=1}^d \psi(X)|X\rangle$, with $\sum_{X=1}^d |\psi(X)|^2 = 1$.

At a high level, a quantum circuit then consists of multiple qudits, and a sequence of quantum gates, which are unitary operations acting on (a subset of) the qudits, with unitarity required to preserve the normalization of the global state of the system. To be more precise, consider a collection of $N$ $d$-dimensional qudits, each described by unit vectors in $\mathcal{H}_i = \mathbb{C}^d$, where $i \in \{1, \ldots, N\}$ indicates a particular qudit. Given such a collection of qudits, the global state of the system is described by a unit vector $|\psi\rangle \in \mathcal{H} \equiv \bigotimes_{i=1}^N \mathcal{H}_i = \mathbb{C}^{d^N}$. In particular, with respect to a fixed ortho-normal basis $\{|X_i\rangle\}_{X=1}^d$ for each sub-system Hilbert space $\mathcal{H}_i$, the global state is specified by $d^N$ complex amplitudes $\psi(X_1, \ldots, X_N)$ - i.e.,

$$|\psi\rangle = \sum_{X_1=1}^d \ldots \sum_{X_N=1}^d \psi(X_1, \ldots, X_N)|X_1\rangle \otimes \ldots \otimes |X_N\rangle, \tag{S3}$$

with the constraint that

$$\langle\psi|\psi\rangle = \sum_{X_1=1}^d \ldots \sum_{X_N=1}^d |\psi(X_1, \ldots, X_N)|^2 = 1. \tag{S4}$$

Note that the set of all amplitudes, which completely defines the state vector $|\psi\rangle$ with respect to this particular basis, is naturally represented as an order-$N$ tensor with $d$-dimensional indices:

$$\psi(X_1, \ldots, X_N) = \begin{array}{c} X_1 \quad\quad X_N \\ \boxed{\phantom{XXXXXX}} \end{array}. \tag{S5}$$

Furthermore, in this diagrammatic notation where legs that join two tensors represent a summation over the corresponding indices of the tensors, the normalization constraint takes the particularly simple form,

$$\langle\psi|\psi\rangle = \begin{array}{c} \boxed{\phantom{XXXX}} \\ \boxed{\phantom{XXXX}} \end{array} = 1, \tag{S6}$$

where the upper tensor is taken to be the complex conjugate of the lower tensor.

Given some initial state of the system, a quantum circuit then consists of a sequence of unitary operations (referred to as *gates*), each of which preserves the normalization of the global state of the system. We are particularly interested in *local* quantum circuits, which consist of unitary operations which act only on a subset of qubits. To be more precise, given some Hilbert space $\mathcal{H}$, let us denote the set of unitary operators acting on elements of $\mathcal{H}$ as $\mathcal{U}(\mathcal{H})$. We will be concerned with 2-local quantum circuits, which consist of unitary operations acting only on pairs of neighbouring qudits - i.e., all gates $U \in \mathcal{U}(\mathcal{H})$ are of the form

$$U = \mathbb{1}_1 \otimes \ldots \otimes \mathbb{1}_{i-1} \otimes U_{i,i+1} \otimes \mathbb{1}_{i+2} \otimes \ldots \otimes \mathbb{1}_N, \tag{S7}$$

for some $i \in \{1, \ldots, N\}$, where $U_{j,j+1} \in \mathcal{U}(\mathcal{H}_j \otimes \mathcal{H}_{j+1})$ and $\mathbb{1}_k$ is the identity operator on $\mathcal{H}_k$.

Let us now consider a quantum circuit in a one-dimensional geometry, consisting of $N$ $d$-dimensional qudits, all initialized in the $|0\rangle$ state vector, to which $D$ layers of 2-local unitary gates are applied. As shown in Equation (S8) below one can write the output state of this circuit as an MPS by first splitting each unitary operator through a singular value decomposition, and then contracting all the resulting tensors as indicated by the dashed boxes. Note that as a result of 2-locality, each unitary operator has rank less than $d^2$, and therefore, the MPS has TT-rank less than $d^{D+1}$.

$$\tag{S8}$$

Finally, given the outcome state vector $|\psi\rangle$ of a quantum circuit, it is necessary to understand the measurement process, via which classical information can be extracted from this state. To this end, we need to understand the Born rule of quantum mechanics. More specifically, in the restricted setting of finite-dimensional Hilbert spaces which we consider here, *observables* correspond to Hermitian operators, and the Born rule states that measurement of an observable $O$ will yield one of the eigenvalues $\lambda_i$ of $O$, with probability $\langle\psi|\Pi_i|\psi\rangle$, where $\Pi_i$ is the projection onto the eigenspace of $\lambda_i$. Note that the normalization of $|\psi\rangle$ is required precisely to allow for this probabilistic interpretation of measurements via the Born rule.

Let us now consider an observable $O$ which is diagonal in the fixed basis we have previously considered (often referred to as the "computational basis"). In this case, we can write

$$O = \sum_{X_1=1}^{d} \ldots \sum_{X_N=1}^{d} \lambda(X_1, \ldots, X_N)|X_1\rangle\langle X_1| \otimes \ldots \otimes |X_N\rangle\langle X_N|, \tag{S9}$$

$$= \sum_{X_1=1}^{d} \ldots \sum_{X_N=1}^{d} \lambda(X_1, \ldots, X_N)\Pi(X_1, \ldots, X_N), \tag{S10}$$

and we find that $P(X_1, \ldots, X_N)$, the probability of obtaining measurement outcome $\lambda(X_1, \ldots, X_N)$, is given by

$$P(X_1, \ldots, X_N) = \langle\psi|\Pi(X_1, \ldots, X_N)|\psi\rangle, \tag{S11}$$

$$= |\psi(X_1, \ldots, X_N)|^2, \tag{S12}$$

$$\tag{S13}$$

As such, we find that measurements of the observable $O$ allow us to sample from the probability mass function $P(X_1, \ldots, X_N) = |\psi(X_1, \ldots, X_N)|^2$, and that when $|\psi\rangle$ is the output state of a 2-local quantum circuit of depth $D$, this probability mass function is exactly a Born machine (where the origin of the name is now clear) of Born-rank $d^{D+1}$. This shows that in this probabilistic modeling approach, local quantum circuits of fixed depth are Born machines of fixed Born-rank.

## 1.3 Local quantum circuits with ancillas are locally purified states

In order to understand the relationship between locally purified states and local quantum circuits with ancillas, it is necessary to understand the effect and formalism of measurements on subsystems.

To this end, consider a one-dimensional array of $2N$ $d$-dimensional qudits, consisting of alternating pairs of *system* and *ancilla* qudits respectively, where each system qudit is a unit vector in $\mathcal{H}_i^{(S)} = \mathbb{C}^d$ spanned by ortho-normal basis $\{|X_i\rangle\}_{X=1}^d$, and each ancilla qudit is a unit vector in $\mathcal{H}_j^{(A)} = \mathbb{C}^\mu$ spanned by $\{|Y_j\rangle\}_{Y=1}^\mu$. The global state vector $|\psi\rangle$ is therefore an element of the Hilbert space $\mathcal{H} = \mathcal{H}_1^{(S)} \otimes \mathcal{H}_1^{(A)} \otimes \ldots \otimes \mathcal{H}_N^{(S)} \otimes \mathcal{H}_N^{(A)}$ - i.e.,

$$|\psi\rangle = \sum_{X_1,\ldots,X_N=1}^{d} \sum_{Y_1,\ldots,Y_N=1}^{\mu} \psi(X_1, Y_1, \ldots, X_N, Y_N)|X_1\rangle \otimes |Y_1\rangle \otimes \ldots \otimes |X_N\rangle \otimes |Y_N\rangle. \tag{S14}$$

As in the previous section, we can consider a 2-local quantum circuit of depth $D$, where all qudits (system and ancilla) are initialized in the $|0\rangle$ state vector, and note that the output state vector $|\psi\rangle$ can again be written as a matrix product state of TT-rank less than $r^{D+1}$, where $r = \min(d, \mu)$.

$$\tag{S15}$$

Now, consider an observable $O$, as in Equation (S10), which is defined only on the system qudits, and is diagonal in the computational basis for this subsystem - i.e.,

$$O = \sum_{X_1=1}^{d} \ldots \sum_{X_N=1}^{d} \lambda(X_1, \ldots, X_N)\Pi(X_1 \ldots, X_N). \tag{S16}$$

The postulates of quantum mechanics state that the probability $P(X_1, \ldots, X_N)$ of measurement outcome $\lambda(X_1, \ldots, X_N)$, when performing a measurement of observable $O$ on the system qudits, is given by

$$P(X_1, \ldots, X_N) = \mathrm{Tr}\big(\rho_S \Pi(X_1, \ldots, X_N)\big), \tag{S17}$$

where $\rho_S$ is the system *density matrix*, given by

$$\rho_S = \mathrm{Tr}_A(|\psi\rangle\langle\psi|), \tag{S18}$$

and where $\mathrm{Tr}_A$ indicates the partial trace over the Hilbert space of all ancilla qudits. Luckily, equations (S17) and (S18) are both easily and concisely expressed in tensor network notation (which is in fact a particularly strong motivation for such a notation). Specifically,

$$\tag{S19}$$

and therefore

$$P(X_1, \ldots, X_N) = \sum_{Y_1,\ldots Y_N=1}^{\mu} |\psi(X_1, Y_1, \ldots, X_N, Y_N)|^2 \tag{S20}$$

$$\tag{S21}$$

As such, we find that measurements of the observable $O$ on the system qudits allow us to sample from the probability mass function (S21), which is precisely a locally purified state of puri-rank $r^{D+1}$, where $r = \min(d, \mu)$, if $|\psi\rangle$ is the output state vector of a 2-local quantum circuit of depth $D$, consisting of $2N$ alternating $d$-dimensional system and $\mu$-dimensional local ancilla qudits.

## 2  Proofs on the expressive power of tensor networks

We provide here proofs for all propositions in Section 5 of the main text. To facilitate ease of presentation and understanding, we restate the propositions here. We begin with the following proposition, concerning inclusions between the sets of probability distributions which can be exactly represented by different tensor-network representations of the same rank.

**Proposition 1.** *For all non-negative tensors $TT\text{-}rank_{\mathbb{R}_{\geq 0}} \geq TT\text{-}rank_{\mathbb{R}}$, $Born\text{-}rank_{\mathbb{R}} \geq Born\text{-}rank_{\mathbb{C}}$, $Born\text{-}rank_{\mathbb{R}} \geq puri\text{-}rank_{\mathbb{R}}$, $Born\text{-}rank_{\mathbb{C}} \geq puri\text{-}rank_{\mathbb{C}}$, $puri\text{-}rank_{\mathbb{R}} \geq puri\text{-}rank_{\mathbb{C}}$, $TT\text{-}rank_{\mathbb{R}_{\geq 0}} \geq puri\text{-}rank_{\mathbb{R}}$, $TT\text{-}rank_{\mathbb{R}} = TT\text{-}rank_{\mathbb{C}}$.*

Proposition 1 is proven via Lemmas 1- 3 below:

**Lemma 1.** *For all non-negative tensors, $TT\text{-}rank_{\mathbb{R}_{\geq 0}} \geq TT\text{-}rank_{\mathbb{R}}$, $Born\text{-}rank_{\mathbb{R}} \geq Born\text{-}rank_{\mathbb{C}}$, $Born\text{-}rank_{\mathbb{R}} \geq puri\text{-}rank_{\mathbb{R}}$, $Born\text{-}rank_{\mathbb{C}} \geq puri\text{-}rank_{\mathbb{C}}$, $puri\text{-}rank_{\mathbb{R}} \geq puri\text{-}rank_{\mathbb{C}}$.*

*Proof.* It is clear that enlarging the set of tensor elements can only reduce the corresponding rank. Moreover a BM is a LPS with purification dimension $\mu = 1$. □

**Lemma 2.** *For all non-negative tensors, $TT\text{-}rank_{\mathbb{R}} = TT\text{-}rank_{\mathbb{C}}$.*

*Proof.* The canonical MPS decomposition of a non-negative tensor can be obtained by successive singular value decompositions [4], and has the same TT-rank as the highest rank across a bipartition. Because the rank of a non-negative matrix is the same over $\mathbb{R}$ or $\mathbb{C}$, $TT\text{-}rank_{\mathbb{R}} = TT\text{-}rank_{\mathbb{C}}$. □

**Lemma 3.** *For all non-negative tensors, $TT\text{-}rank_{\mathbb{R}_{\geq 0}} \geq puri\text{-}rank_{\mathbb{R}}$*

*Proof.* Let us denote the non-negative tensors of an MPS of $TT\text{-}rank_{\mathbb{R}_{\geq 0}} = r$ as $A_i$. We define a $LPS_{\mathbb{R}}$ of purification index of size $\mu = r^2$ with the tensors

$$B_{1,X_1}^{\beta_1,\alpha_1} = \delta_{\alpha_1,\beta_1} \sqrt{A_{1,X_1}^{\alpha_1}}, \tag{S22}$$

$$B_{N,X_N}^{\beta_N,\alpha_{N-1}} = \delta_{\alpha_{N-1},\beta_N} \sqrt{A_{N,X_N}^{\alpha_{N-1}}}, \tag{S23}$$

$$B_{i,X_i}^{\beta_i,\alpha_{i-1},\alpha i} = \delta_{\alpha_{i-1}r+\alpha_i,\beta_i} \sqrt{A_{i,X_i}^{\beta_i,\alpha_{i-1},\alpha i}}. \tag{S24}$$

We now observe that

$$\sum_{\beta_i} B_{i,X_i}^{\beta_i,\alpha_{i-1},\alpha i} \overline{B}_{i,X_i}^{\beta_i,\alpha'_{i-1},\alpha' i} = \sum_{\beta_i} \delta_{\alpha_{i-1}r+\alpha_i,\beta_i} \delta_{\alpha'_{i-1}r+\alpha'_i,\beta_i} \sqrt{A_{i,X_i}^{\beta_i,\alpha_{i-1},\alpha i}} \sqrt{A_{i,X_i}^{\beta_i,\alpha'_{i-1},\alpha' i}} \tag{S25}$$

$$= \delta_{\alpha_{i-1},\alpha'_{i-1}} \delta_{\alpha_i,\alpha'_i} A_{i,X_i}^{\alpha_{i-1},\alpha i}, \tag{S26}$$

or equivalently, in graphical notation, that

$$\tag{S27}$$

Therefore, this LPS defines the same tensor as the original MPS and has $puri\text{-}rank_{\mathbb{R}} = r$. □

We now turn to Proposition 2, showing that all inequalities given in Proposition 1 can in fact be strict, and that for all other pairs of representations there exist probability distributions (non-negative tensors) showing that neither rank can always be lower than the other.

**Proposition 2.** *The ranks of all introduced tensor-network representations satisfy the properties contained in Table 1. Specifically, denoting by $r_{row}$ ($r_{column}$) the rank appearing in the row (column), $<$ indicates that there exists a tensor satisfying $r_{row} < r_{column}$ and $<, >$ indicates that there exists both a tensor satisfying $r_{row} < r_{column}$ and another tensor satisfying $r_{column} > r_{row}$.*

Again, we prove Proposition 2 via Lemmas 4-9, each of which addresses a subset of the entries in Table 1. The particular entry addressed by a specific lemma is indicated by $[k,l]_{>}$ or $[k,l]_{<}$, where $k$ denotes the row and $l$ the column of Table 1, and the subscript $<$ ($>$) is used to indicate a specific case. Note also that a tensor providing a proof for entry $[k,l]_{>}$ ($[k,l]_{<}$) provides also a proof for entry $[l,k]_{<}$ ($[l,k]_{>}$).

Table 1: Results of Proposition 2

| | TT-rank$_\mathbb{R}$ | TT-rank$_{\mathbb{R}_{\geq 0}}$ | Born-rank$_\mathbb{R}$ | Born-rank$_\mathbb{C}$ | puri-rank$_\mathbb{R}$ | puri-rank$_\mathbb{C}$ |
|---|---|---|---|---|---|---|
| TT-rank$_\mathbb{R}$ | $=$ | $<$ | $<,>$ | $<,>$ | $<,>$ | $<,>$ |
| TT-rank$_{\mathbb{R}_{\geq 0}}$ | $>$ | $=$ | $<,>$ | $<,>$ | $>$ | $>$ |
| Born-rank$_\mathbb{R}$ | $<,>$ | $<,>$ | $=$ | $>$ | $>$ | $>$ |
| Born-rank$_\mathbb{C}$ | $<,>$ | $<,>$ | $<$ | $=$ | $<,>$ | $>$ |
| puri-rank$_\mathbb{R}$ | $<,>$ | $<$ | $<$ | $<,>$ | $=$ | $>$ |
| puri-rank$_\mathbb{C}$ | $<,>$ | $<$ | $<$ | $<$ | $<$ | $=$ |

**Lemma 4** ([1, 2])**.** *There exists a non-negative matrix A with TT-rank$_\mathbb{R}$ < TT-rank$_{\mathbb{R}_{\geq 0}}$.*

*Proof.* Consider the matrix $A = \begin{pmatrix} 0 & 1 & 1 & 0 \\ 0 & 0 & 1 & 1 \\ 1 & 0 & 0 & 1 \\ 1 & 1 & 0 & 0 \end{pmatrix}$. $A$ has TT-rank$_\mathbb{R} = 3$ and TT-rank$_{\mathbb{R}_{\geq 0}} = 4$ [5]. $\square$

**Lemma 5** ([1, 3]$_<$, [2, 3]$_<$, [3, 4], [3, 5], [3, 6])**.** *There exists a non-negative matrix B with TT-rank$_\mathbb{R}$ < Born-rank$_\mathbb{R}$, TT-rank$_{\mathbb{R}_{\geq 0}}$ < Born-rank$_\mathbb{R}$ and Born-rank$_\mathbb{R}$ > puri-rank$_\mathbb{R}$.*

*Proof.* Consider the matrix $B = \begin{pmatrix} 2 & 1 & 1 \\ 1 & 0 & 1 \\ 1 & 1 & 0 \end{pmatrix}$. $B$ has TT-rank$_\mathbb{R} = 2$ and TT-rank$_{\mathbb{R}_{\geq 0}} = 2$. Moreover the square root element-wise of $B$ is $\begin{pmatrix} \sqrt{2} & 1 & 1 \\ 1 & 0 & 1 \\ 1 & 1 & 0 \end{pmatrix}$, which has rank 3, as well as all square roots obtained by changing signs of each element, so Born-rank$_\mathbb{R} = 3$. On the other hand it is also possible to write $B$ as the absolute value squared of $\begin{pmatrix} 1+i & 1 & 1 \\ 1 & 0 & 1 \\ i & 1 & 0 \end{pmatrix}$, which has rank 2, so Born-rank$_\mathbb{C} = 2$. Furthermore, from Proposition 1 and the fact that TT-rank$_{\mathbb{R}_{\geq 0}} = 2$, we have that puri-rank$_{\mathbb{R}/\mathbb{C}} \leq 2$. $\square$

**Lemma 6** ([2, 5], [2, 6], [2, 3]$_>$, [2, 4]$_>$, [1, 3]$_>$, [1, 4]$_>$, [1, 5]$_>$, [1, 6]$_>$)**.** *There exists a non-negative matrix C with TT-rank$_{\mathbb{R}_{\geq 0}}$ > puri-rank$_\mathbb{R}$, TT-rank$_{\mathbb{R}_{\geq 0}}$ > puri-rank$_\mathbb{C}$, TT-rank$_{\mathbb{R}_{\geq 0}}$ > Born-rank$_\mathbb{R}$, TT-rank$_{\mathbb{R}_{\geq 0}}$ > Born-rank$_\mathbb{C}$, TT-rank$_\mathbb{R}$ > Born-rank$_\mathbb{R}$, TT-rank$_\mathbb{R}$ > Born-rank$_\mathbb{C}$, TT-rank$_\mathbb{R}$ > puri-rank$_\mathbb{R}$ and TT-rank$_\mathbb{R}$ > puri-rank$_\mathbb{C}$.*

*Proof.* Consider the matrix $C = \begin{pmatrix} 4 & 1 & 1 \\ 1 & 0 & 1 \\ 1 & 1 & 0 \end{pmatrix}$. $C$ has TT-rank$_{\mathbb{R}_{\geq 0}} =$ TT-rank$_\mathbb{R} = 3$, but the square root is matrix $B$ from Lemma 5, of rank 2, so Born-rank$_\mathbb{R} =$ Born-rank$_\mathbb{C} = 2$. Again, from Proposition 1 and the fact that Born-rank$_\mathbb{R} = 2$ we have that puri-rank$_{\mathbb{R}/\mathbb{C}} \leq 2$. $\square$

**Lemma 7** ([1, 4]$_<$, [1, 5]$_<$, [1, 6]$_<$)**.** *There exists a non-negative matrix D with TT-rank$_\mathbb{R}$ < Born-rank$_\mathbb{C}$, TT-rank$_\mathbb{R}$ < puri-rank$_\mathbb{R}$ and TT-rank$_\mathbb{R}$ < puri-rank$_\mathbb{C}$.*

*Proof.* Consider $a = (1 + \sqrt{5})/2$ and define $D = \begin{pmatrix} 0 & 1 & a & 1 & 0 \\ 0 & 0 & 1 & a & 1 \\ 1 & 0 & 0 & 1 & a \\ a & 1 & 0 & 0 & 1 \\ 1 & a & 1 & 0 & 0 \end{pmatrix}$. $D$ is the slack matrix of a regular pentagon, and has TT-rank$_{\mathbb{R}_{\geq 0}} = 5$ while TT-rank$_\mathbb{R} = 3$. $D$ has puri-rank$_\mathbb{R} = 4$ and puri-rank$_\mathbb{C} = 4$, as proven in ref. [6]. By Proposition 1 and the fact that puri-rank$_\mathbb{C} = 4$ we have that Born-rank$_\mathbb{C} \geq 4$. $\square$

**Lemma 8** ([5, 6], [4, 5]$_<$)**.** *There exists a non-negative matrix E with puri-rank$_\mathbb{R}$ > puri-rank$_\mathbb{C}$ and Born-rank$_\mathbb{C}$ < puri-rank$_\mathbb{R}$.*

*Proof.* Consider the matrix $E = \begin{pmatrix} 0 & 1 & 1 & 1 \\ 1 & 0 & 1 & 1 \\ 1 & 1 & 0 & 1 \\ 1 & 1 & 1 & 0 \end{pmatrix}$. $E$ can be written as the product of $\begin{pmatrix} 1 & 0 \\ 0 & 1 \\ 1 & -1 \\ 1 & e^{2i\pi/3} \end{pmatrix}$ and

$\begin{pmatrix} 0 & 1 & 1 & 1 \\ 1 & 0 & 1 & -e^{-2i\pi/3} \end{pmatrix}$, which shows that Born-rank$_{\mathbb{C}} \leq 2$, and therefore, puri-rank$_{\mathbb{C}} \leq 2$ by Proposition 1. Bounds on the real positive semidefinite rank imply that it is equal to 3 [7]. $\qquad\square$

**Lemma 9** ([4, 6], [2, 4]$_<$, [4, 5]$_>$). *There exists a non-negative matrix $F$ with puri-rank$_{\mathbb{C}} <$ Born-rank$_{\mathbb{C}}$, TT-rank$_{\mathbb{R}\geq 0} <$ Born-rank$_{\mathbb{C}}$ and Born-rank$_{\mathbb{C}} >$ puri-rank$_{\mathbb{R}}$.*

*Proof.* Consider the matrix $F = \begin{pmatrix} 1 & 0 & 0 & 1 & 1 & 0 & 1 \\ 0 & 1 & 0 & 0 & 1 & 1 & 1 \\ 0 & 0 & 1 & 1 & 0 & 1 & 1 \\ 1 & 0 & 1 & 2 & 1 & 1 & 2 \\ 1 & 1 & 0 & 1 & 2 & 1 & 2 \\ 0 & 1 & 1 & 1 & 1 & 2 & 2 \\ 1 & 1 & 1 & 2 & 2 & 2 & 3 \end{pmatrix}$. $F$ has TT-rank$_{\mathbb{R}} = 3$ and is equal to the

product of $\begin{pmatrix} 0 & 0 & 1 \\ 1 & 0 & 0 \\ 0 & 1 & 0 \\ 0 & 1 & 1 \\ 1 & 0 & 1 \\ 1 & 1 & 0 \\ 1 & 1 & 1 \end{pmatrix}$ and its transpose, so $F$ has TT-rank$_{\mathbb{R}\geq 0} = 3$. In addition, we now prove that $F$ has

Born-rank$_{\mathbb{C}} \geq 4$. To this end, consider a complex Hadamard square root of matrix $F$ given by

$$\sqrt{F} = \begin{pmatrix} e^{i\phi_1} & 0 & 0 & e^{i\phi_2} & e^{i\phi_3} & 0 & e^{i\phi_4} \\ 0 & e^{i\phi_5} & 0 & 0 & e^{i\phi_6} & e^{i\phi_7} & e^{i\phi_8} \\ 0 & 0 & e^{i\phi_9} & e^{i\phi_{10}} & 0 & e^{i\phi_{11}} & e^{i\phi_{12}} \\ e^{i\phi_{13}} & 0 & e^{i\phi_{14}} & \sqrt{2}e^{i\phi_{15}} & e^{i\phi_{16}} & e^{i\phi_{17}} & \sqrt{2}e^{i\phi_{18}} \\ e^{i\phi_{19}} & e^{i\phi_{20}} & 0 & e^{i\phi_{21}} & \sqrt{2}e^{i\phi_{22}} & e^{i\phi_{23}} & \sqrt{2}e^{i\phi_{24}} \\ 0 & e^{i\phi_{25}} & e^{i\phi_{26}} & e^{i\phi_{27}} & e^{i\phi_{28}} & \sqrt{2}e^{i\phi_{29}} & \sqrt{2}e^{i\phi_{30}} \\ e^{i\phi_{31}} & e^{i\phi_{32}} & e^{i\phi_{33}} & \sqrt{2}e^{i\phi_{34}} & \sqrt{2}e^{i\phi_{35}} & \sqrt{2}e^{i\phi_{36}} & \sqrt{3}e^{i\phi_{37}} \end{pmatrix}, \qquad \text{(S28)}$$

where the $\phi_i$ are real parameters. We will prove that the rank of $\sqrt{F}$ is at least 4. First observe that the rank is invariant under multiplication of a row or a column by a phase. By performing such operations in the right order, we obtain that the rank of $\sqrt{F}$ is the same as the rank of a matrix

$$M = \begin{pmatrix} 1 & 0 & 0 & 1 & e^{i\phi_1} & 0 & 1 \\ 0 & 1 & 0 & 0 & 1 & e^{i\phi_2} & 1 \\ 0 & 0 & 1 & e^{i\phi_3} & 0 & 1 & 1 \\ e^{i\phi_4} & 0 & e^{i\phi_5} & \sqrt{2} & e^{i\phi_6} & e^{i\phi_7} & \sqrt{2}e^{i\phi_8} \\ e^{i\phi_9} & e^{i\phi_{10}} & 0 & e^{i\phi_{11}} & \sqrt{2} & e^{i\phi_{12}} & \sqrt{2}e^{i\phi_{13}} \\ 0 & e^{i\phi_{14}} & e^{i\phi_{15}} & e^{i\phi_{16}} & e^{i\phi_{17}} & \sqrt{2} & \sqrt{2}e^{i\phi_{18}} \\ e^{i\phi_{19}} & e^{i\phi_{20}} & e^{i\phi_{21}} & \sqrt{2}e^{i\phi_{22}} & \sqrt{2}e^{i\phi_{23}} & \sqrt{2}e^{i\phi_{24}} & \sqrt{3} \end{pmatrix}, \qquad \text{(S29)}$$

with new real parameters $\phi_i$ (defined modulo $2\pi$). We will prove that such a matrix has always rank at least 4. It is clear that the first three rows are independent, so the rank is at least 3. Now suppose that the rank is 3, the rows 4 to 7 are therefore complex linear combinations of the first 3 rows. Let us write such a linear combination for row 4:

$$(4) = \alpha(1) + \beta(2) + \gamma(3). \qquad \text{(S30)}$$

The first columns imply that $\alpha = e^{i\phi_4}$, $\beta = 0$ and $\gamma = e^{i\phi_5}$. Moreover we have

$$e^{i\phi_4} + e^{i\phi_5}e^{i\phi_3} = \sqrt{2}, \qquad \text{(S31)}$$

$$e^{i\phi_4} + e^{i\phi_5} = \sqrt{2}e^{i\phi_8}. \qquad \text{(S32)}$$

Let us take the absolute value squared of these equations, we obtain

$$2 + 2\cos(\phi_4 - \phi_5 - \phi_3) = 2, \qquad \text{(S33)}$$

$$2 + 2\cos(\phi_4 - \phi_5) = 2. \qquad \text{(S34)}$$

Therefore, $\cos(\phi_4 - \phi_5 - \phi_3) = \cos(\phi_4 - \phi_5) = 0$, which implies that $\phi_4 - \phi_5 = \pm\pi/2$ and $\phi_3 = 0$ or $\pi$, so that $e^{i\phi_3} = \pm 1$. By similarly writing that row 5 and 6 are linear combinations of the first three rows, we obtain

by symmetry that $e^{i\phi_1} = \pm 1$ and $e^{i\phi_2} = \pm 1$. Let us now show that the last row cannot be written as a linear combination of the first three rows. Suppose this is the case, so that

$$(7) = \alpha(1) + \beta(2) + \gamma(3). \tag{S35}$$

Then the first columns imply that $\alpha = e^{i\phi_{19}}$, $\beta = e^{i\phi_{20}}$ and $\gamma = e^{i\phi_{21}}$. We know that $e^{i\phi_1} = \pm 1$, $e^{i\phi_2} = \pm 1$ and $e^{i\phi_3} = \pm 1$. We then have

$$e^{i\phi_{19}} \pm e^{i\phi_{21}} = \sqrt{2}e^{i\phi_{22}}, \tag{S36}$$

$$\pm e^{i\phi_{19}} + e^{i\phi_{20}} = \sqrt{2}e^{i\phi_{23}}, \tag{S37}$$

$$\pm e^{i\phi_{20}} + e^{i\phi_{21}} = \sqrt{2}e^{i\phi_{24}}. \tag{S38}$$

From this we obtain, by taking the absolute value squared,

$$\cos(\phi_{19} - (\phi_{21} \pm \pi)) = 0, \tag{S39}$$
$$\cos((\phi_{19} \pm \pi) - \phi_{20}) = 0, \tag{S40}$$
$$\cos((\phi_{20} \pm \pi) - \phi_{21}) = 0, \tag{S41}$$

which implies

$$\phi_{19} - \phi_{21} = \pm\pi/2, \tag{S42}$$
$$\phi_{19} - \phi_{20} = \pm\pi/2, \tag{S43}$$
$$\phi_{20} - \phi_{21} = \pm\pi/2, \tag{S44}$$

which is impossible. We therefore conclude that $M$, and thus also $\sqrt{F}$, has rank at least 4. $\qquad\square$

Before continuing to Proposition 3, it is interesting to note that the proofs of Lemma's 4 - 9 all involve lower-bounding a given rank, and that this problem may be cast into the form of a polynomial optimization problem, for which hierarchies of semi-definite relaxations are available [8]. For example, the non-negative rank TT-rank$_{\mathbb{R}\geq 0}$ can for a given $d \times d$-matrix $T$ be computed via the minimization problem

$$\min \|T - C\|_2 \tag{S45}$$

subject to $C = AB$, where $A$ and $B$ are $d \times k$ and $k \times d$ matrices with non-negative entries, respectively. A hierarchy of convex relaxations can then be used to provide increasingly better approximations to the optimal solution, and Kuhn-Tucker conditions can be made use of to check for global optimality of a solution. In practice the required relaxations can soon become infeasibly large, however this strategy is worth noting as a potentially interesting tool, particularly for the complex Hadamard square root rank.

Finally, we move onto the proof of Proposition 3, addressing the question of the overheads required to exactly represent a tensor network representation of a given rank with an alternative representation.

**Proposition 3.** *The ranks of all introduced tensor-network representations satisfy the relationships without asterisk contained in Table 2. A function $g(x)$ denotes that for all non-negative tensors $r_{row} \leq g(r_{column})$. "No" indicates that there exists a family of probability distributions of increasing $N$ with $d = 2$ and $r_{column}$ constant, but such that $r_{row}$ goes to infinity, i.e., that no such function can exist.*

Table 2: Results of Proposition 3.

| | TT-rank$_{\mathbb{R}}$ | TT-rank$_{\mathbb{R}\geq 0}$ | Born-rank$_{\mathbb{R}}$ | Born-rank$_{\mathbb{C}}$ | puri-rank$_{\mathbb{R}}$ | puri-rank$_{\mathbb{C}}$ |
|---|---|---|---|---|---|---|
| TT-rank$_{\mathbb{R}}$ | = | $\leq x$ | $\leq x^2$ | $\leq x^2$ | $\leq x^2$ | $\leq x^2$ |
| TT-rank$_{\mathbb{R}\geq 0}$ | No | = | No | No | No | No |
| Born-rank$_{\mathbb{R}}$ | No | No | = | No | No | No |
| Born-rank$_{\mathbb{C}}$ | No | No* | $\leq x$ | = | No* | No* |
| puri-rank$_{\mathbb{R}}$ | No | $\leq x$ | $\leq x$ | $\leq 2x$ | = | $\leq 2x$ |
| puri-rank$_{\mathbb{C}}$ | No | $\leq x$ | $\leq x$ | $\leq x$ | $\leq x$ | = |

Once again, it is convenient to prove Proposition 3 via a series of lemmas. However, note first that all entries of Table 2 containing the function $g(x) = x$ follow straightforwardly from Proposition 1. Given this, we begin with Lemmas 10 and 11 addressing the remaining entries of Table 2 for which explicit functions can be found:

**Lemma 10.** *For all non-negative tensors TT-rank$_{\mathbb{R}} \leq$ (puri-rank$_{\mathbb{C}}$)$^2$, therefore, also TT-rank$_{\mathbb{R}} \leq$ (puri-rank$_{\mathbb{R}}$)$^2$, TT-rank$_{\mathbb{R}} \leq$ (Born-rank$_{\mathbb{R}}$)$^2$ and TT-rank$_{\mathbb{R}} \leq$ (Born-rank$_{\mathbb{C}}$)$^2$.*

*Proof.* Consider an LPS$_{\mathbb{C}}$ of puri-rank$_{\mathbb{C}} = r$. Let us denote the tensors defining this LPS as $A_{i,X_i}^{\beta_i,\alpha_{i-1},\alpha_i}$. Define new tensors $B_{i,X_i}^{\alpha_{i-1},r+\alpha'_{i-1},\alpha_i,r+\alpha'_i} = \sum_{\beta_i} A_{i,X_i}^{\beta_i,\alpha_{i-1},\alpha_i} \overline{A}_{i,X_i}^{\beta_i,\alpha'_{i-1},\alpha'_i}$. As shown in Equation (S46),

these tensors define an $\text{MPS}_\mathbb{R}$ of $\text{TT-rank}_\mathbb{R} = r^2$ corresponding to the same probability mass function as the original LPS.

$$(\text{S46})$$

$\square$

**Lemma 11.** *For all non-negative tensors $\text{puri-rank}_\mathbb{R} \leq 2\text{puri-rank}_\mathbb{C}$ and $\text{puri-rank}_\mathbb{R} \leq 2\text{Born-rank}_\mathbb{C}$.*

*Proof.* Consider an $\text{LPS}_\mathbb{C}$ with purification dimension equal to $\mu$ and $\text{puri-rank}_\mathbb{C}$ equal to $r$, constructed from tensors $A_i$. If we fix $i$ and $X_i$, $A_i$ is an $r \times \mu$ matrix for $i = 1$ or $i = N$ and an order-3 tensor of size $r \times r \times \mu$ otherwise. Now define new tensors by blocks as

$$B_1 = \begin{pmatrix} \text{Re}(A_1) & -\text{Im}(A_1) \\ \text{Im}(A_1) & \text{Re}(A_1) \end{pmatrix}, \quad B_N = \begin{pmatrix} \text{Re}(A_N) & \text{Im}(A_1) \\ -\text{Im}(A_N) & \text{Re}(A_N) \end{pmatrix}, \qquad (\text{S47})$$

$$B_i = \qquad\qquad\qquad\qquad\qquad\qquad\qquad , \quad \forall i \in \{2, \dots, N-1\}. \qquad (\text{S48})$$

Here $B_i$ is an order-3 tensor defined by blocks, where each block has dimension $r \times r \times \mu$. These tensors define a real LPS with purification dimension equal to $2\mu$ and $\text{puri-rank}_\mathbb{R}$ equal to $2r$ which represents the same probability mass function as the original complex LPS. Applying this result when $\mu = 1$ shows that $\text{puri-rank}_\mathbb{R} \leq 2\text{Born-rank}_\mathbb{C}$. $\square$

We now move on to the proofs for the "No" entries of Table 2. As discussed in the main text, for each "No" entry the strategy is to first prove the existence of a family of non-negative matrices (probability distributions over two discrete random variables) with the property that $r_{\text{column}}$ remains constant with respect to the dimension of the matrix, while $r_{\text{row}}$ grows. To this end, consider lemmas 12-15, each of which addresses a specific entry of Table 2, for the restricted case of only two random variables:

**Lemma 12** ([6, 1])**.** *There exists a family of non-negative matrices, of increasing dimension d, with rank equal to 3 and $\text{puri-rank}_\mathbb{C} \geq \Omega(\log d)$.*

*Proof.* Slack matrices of regular $n$-gons in the plane have and rank 3 but $\text{puri-rank}_\mathbb{C} \geq \Omega(\log n)$ [9]. $\square$

**Lemma 13** ([3, 2])**.** *There exists a family of non-negative matrices, of increasing dimension d, with $\text{TT-rank}_{\mathbb{R}_{\geq 0}} = 2$ and $\text{Born-rank}_\mathbb{R} = d$.*

*Proof.* Consider a sequence of integers $n_i$ such that $2n_i - 1$ is the $i$-th prime. Define the primes matrices $K_{i,j} = n_i + n_j - 1$. $K$ has rank 2 and non-negative rank 2. It was shown by induction in ref. [7] that the real square root rank of $K$ is full. $\square$

**Lemma 14** ([2, 3])**.** *There exists a family of non-negative matrices, of increasing dimension d, with $\text{Born-rank}_\mathbb{R} = 2$ and $\text{TT-rank}_{\mathbb{R}_{\geq 0}} \geq \log_2 d$.*

*Proof.* Consider the linear Euclidean distance matrices defined as $M_{i,j} = (j-i)^2$. $M$ is the element-wise square of a matrix with elements equal to $i - j$, so has real square root rank equal to 2. Moreover, it was shown in ref. [7] that $M$ has non-negative rank at least $\log_2 d$. $\square$

**Lemma 15** ([3, 4])**.** *There exists a family of non-negative matrices, with increasing dimension d, with $\text{Born-rank}_\mathbb{C} = 2$ and $\text{Born-rank}_\mathbb{R} = d$.*

*Proof.* The prime matrices $K$ introduced in Lemma 13 have full real square root rank. They can be written as the absolute value squared element-wise of a matrix $M_{i,j} = \sqrt{n_i} + i\sqrt{n_j - 1}$. $M$ has rank 2, so the complex square root rank of $K$ is 2. $\qquad\square$

Before continuing, note that all the remaining "No" entries not explicitly covered by Lemmas 12-15 in fact follow directly from these lemmas when combined with Proposition 1 (this is made explicit shortly, in Propositions 4-7).

Given these families of probability distributions over two random variables, we now extend these results to the case of probability mass functions over many variables of small dimension. As discussed in the main text, for a particular $[row, column]$ entry, the strategy is to start with a matrix $M$ of size $2^N \times 2^N$ such that $r_{\text{column}}$ is constant with respect to $N$, while $r_{\text{row}}$ grows. Via an "unfolding" technique, applied to the *column* decomposition of $M$, we then show that there exists a non-negative tensor with $2N$ two-dimensional indices such that (a) the tensor rank corresponding to the column is equal to $r_{\text{column}}$ and (b) the matrix M is a reshaping of the central bipartition of the tensor. As a result of (b) it then follows that the tensor rank corresponding to the row is lower bounded by $r_{\text{row}}$, therefore extending the separation from the case of two-variables to the case of many variables of small dimension.

$$ \text{(S49)} $$

More generally, we can write any $N \times N$ matrix $M$ as a submatrix of a $2^N \times 2^N$ matrix for which we can apply the previous idea. In this case $M$ is a submatrix of the central bipartition of the obtained tensor over $2N$ binary variables,

$$ \text{(S50)} $$

The "unfolding" technique upon which this proof strategy relies is formalized by Lemma 16:

**Lemma 16.** *Consider a non-negative matrix $M$ of TT-rank$_{\mathbb{R}_{\geq 0}}$ (resp. TT-rank$_{\mathbb{R}}$, Born-rank$_{\mathbb{R}}$, Born-rank$_{\mathbb{C}}$) equal to $r$ and size $N \times N$. Then there exists an MPS$_{\mathbb{R}_{\geq 0}}$ (resp. MPS$_{\mathbb{R}}$, BM$_{\mathbb{R}}$, BM$_{\mathbb{C}}$) over $2N$ binary variables with TT-rank$_{\mathbb{R}_{\geq 0}}$ (resp. TT-rank$_{\mathbb{R}}$, Born-rank$_{\mathbb{R}}$, Born-rank$_{\mathbb{C}}$) equal to $r$ such that $M$ is a submatrix of the central bipartition of the resulting tensor.*

*Proof.* Let us first prove the case where $M$ has TT-rank$_{\mathbb{R}}$ $r$. In this case we can write

$$ M_{i,j} = \sum_{\alpha=1}^{r} E_{i,\alpha} F_{\alpha,j}, \tag{S51} $$

where $E$ and $F$ are real matrices.

We now define the appropriate MPS of TT-rank $r$ by direct specification of its tensors. Specifically, define the boundary tensors for site one ($2N$) such that the row (column) vector in the zero-index is a vector of ones and the row (column) vector in the one-index is the first row (last column) of $E$ ($F$), i.e.,

$$ A_{1,0}^{\alpha} = 1, \quad A_{1,1}^{\alpha} = E_{1,\alpha}, \tag{S52} $$
$$ A_{2N,0}^{\alpha} = 1, \quad A_{2N,1}^{\alpha} = F_{\alpha,N}. \tag{S53} $$

We then define the bulk tensors such that left (right) of the central bipartition the matrix in the zero-index is the identity matrix, while the matrix in the one-index is diagonal with a row (column) of $E$ ($F$) on the diagonal, i.e., for all $n$ in $\{2, \ldots, N\}$,

$$ A_{n,0}^{\alpha,\beta} = \delta_{\alpha,\beta}, \quad A_{n,1}^{\alpha,\beta} = \delta_{\alpha,\beta} E_{n,\alpha}, \tag{S54} $$
$$ A_{n-1+N,0}^{\alpha,\beta} = \delta_{\alpha,\beta}, \quad A_{n-1+N,1}^{\alpha,\beta} = \delta_{\alpha,\beta} F_{\alpha,n}. \tag{S55} $$

This MPS defines a tensor over $2N$ variables.

$$ \text{(S56)} $$

Define $R$ the $2^N \times 2^N$ matrix corresponding to a reshaping as a matrix of this tensor across the central bipartition. Consider $R_{0\cdots010\cdots0,0\cdots010\cdots0}$, where the variables are all 0 except a 1 in position $i \leq d$ and a 1 in position $j \geq d+1$, then $R_{0\cdots010\cdots0,0\cdots010\cdots0} = \sum_{\alpha=1}^{r} E_{i,\alpha} F_{\alpha,j} = M_{i,j}$. Therefore, $M$ is a submatrix of $R$, up to a reshaping of $R$ as a matrix. The exact same proof can be done if $M$ has TT-rank$_{\mathbb{R}_{\geq 0}} = r$.

If $M$ has Born-rank$_{\mathbb{R}}$ (resp. Born-rank$_{\mathbb{C}}$) $r$, apply the previous result to a real (resp. complex) element-wise square root of M of rank $r$. This leads to an MPS for which the square root of $M$ is a submatrix of the central bipartition of the MPS. Therefore, $M$ is a submatrix of the central bipartition of the tensor obtained from the corresponding BM, which is the square of this MPS. □

Via the strategy discussed above - i.e., applying the unfolding to the matrix examples in Lemmas 12 -15 (with respect to the decomposition for which the corresponding rank remains constant) - we are able to use Lemma 16 to leverage the matrix families from Lemmas 12-15 into families of probability distributions over $N$ random variables of dimension 2, which prove all the "No" entries in Table 2 (when combined with Proposition 1). Note that the entries "No$^*$" remain conjectures. The existence of a family of matrices of constant non-negative rank but unbounded complex Hadamard square root rank, together with Lemma 16, would prove these entries.

While Lemma 16 provides an explicit construction for the "No" entries in Table 2, the separations it provides are not optimal, since it is sometimes possible to unfold a $2^N \times 2^N$ matrix into a tensor network of only $2N$ variables, as in Equation (S49), rather than into a tensor network of $2(2^N)$ variables as is done in Lemma 16. For this reason we provide more detailed proofs for the explicit asymptotics of the relevant separations, all of which use a similar strategy, but some of which use alternative unfolding techniques. These results are stated here as Propositions 4-7 to reflect their discussion in the main text. In particular, in order to obtain the asymptotic separations given in Propositions 4-6, it is necessary to use alternative unfolding techniques to the one presented in Lemma 16. Once again, the propositions are labelled by the the specific cases of Table 2 which they address.

**Proposition 4** ([6, 1], [5, 1], [4, 1], [3, 1], [2, 1]). *There exists a family of non-negative tensors over $2N$ binary variables with constant TT-rank$_{\mathbb{R}}$=3 but with puri-rank$_{\mathbb{C}} = \Omega(N)$, and hence also puri-rank$_{\mathbb{C}}$, Born-rank$_{\mathbb{R}/\mathbb{C}}$ and TT-rank$_{\mathbb{R}_{\geq 0}} \geq \Omega(N)$.*

*Proof.* The result for the case [6, 1] has already been proven in ref. [10]. Note that the remaining cases then follow from Proposition 1. □

**Proposition 5** ([3, 2], [3, 5], [3, 6]). *There exists a family of non-negative tensors over $2N$ binary variables with constant TT-rank$_{\mathbb{R}_{\geq 0}} = 2$ (and hence also puri-rank$_{\mathbb{R}/\mathbb{C}} = 2$) but with Born-rank$_{\mathbb{R}} \geq \pi(2^{N+1})$, where $\pi(x)$ is the number of prime numbers up to $x$, which asymptotically satisfies $\pi(x) \sim x/\log(x)$.*

*Proof.* Consider the $2^N \times 2^N$ matrix with entries $M_{i,j} = i + j$. A submatrix of $M$ is the prime matrix $K$ defined in Lemma 13 of size $\pi(2^{N+1})$, where $\pi(x)$ is the number of prime numbers lower than $x$. Let us show that we can define an MPS$_{\mathbb{R}_{\geq 0}}$ of TT-rank$_{\mathbb{R}_{\geq 0}} = 2$ such that $M$ is the central bipartition of the resulting tensor. Let us first define the matrices

$$P_N = \begin{pmatrix} 1 & 1 \\ 1 & 2 \\ \vdots & \vdots \\ 1 & 2^N \end{pmatrix}, Q_N = \begin{pmatrix} 1 & 2 & \cdots & 2^N \\ 1 & 1 & \cdots & 1 \end{pmatrix}, \tag{S57}$$

and observe that $M = P_N Q_N$,

$$M = \boxed{P_N} \!\!-\!\! \boxed{Q_N}. \tag{S58}$$

Now consider the tensors

$$A_{1,0} = \begin{pmatrix} 1 & 1 \end{pmatrix}, \quad A_{1,1} = \begin{pmatrix} 1 & 2 \end{pmatrix}, \tag{S59}$$

$$\forall n \in \{2, \ldots, N\}, \quad A_{n,0} = \begin{pmatrix} 1 & 0 \\ 0 & 1 \end{pmatrix}, \quad A_{n,1} = \begin{pmatrix} 1 & 2^n \\ 0 & 1 \end{pmatrix}, \tag{S60}$$

and build an MPS by contracting tensors $A_1$ to $A_N$, as

 (S61)

 (S62)

We obtain a tensor $T_N$ with $N$ open indices corresponding to $N$ binary variables and an extra virtual index of dimension 2. If we reshape this tensor as a $2^N \times 2$ matrix we obtain matrix $P_N$. In the same way we can obtain an MPS of non-negative tensors such that contracting $N$ sites gives a tensor that can be reshaped as $Q_N$. Contracting the two extra virtual indices between the two MPS, we finally obtain an $\text{MPS}_{\mathbb{R}_{\geq 0}}$ over $2N$ variables such that $M$ is the central bipartition of the resulting tensor. Suppose that there is a $\text{BM}_{\mathbb{R}}$ defining the same probability mass function over $2N$ variables, then it has Born-rank$_{\mathbb{R}}$ larger or equal to the square root rank of $M$, which is larger than the square root rank of $K$, which is $\pi(2^{N+1})$. This proves the case $[3, 2]$, the remaining cases follow directly from Proposition 1. $\qquad\square$

**Proposition 6** ($[2, 3]$, $[2, 4]$, $[2, 5]$, $[2, 6]$ )**.** *There exists a family of non-negative tensors over $2N$ binary variables with constant Born-rank$_{\mathbb{R}} = 2$ (and hence also constant Born-rank$_{\mathbb{C}}$ and puri-rank$_{\mathbb{R}/\mathbb{C}}$) that have TT-rank$_{\mathbb{R}_{\geq 0}} \geq N$.*

*Proof.* Consider the linear Euclidean matrices (see Lemma 14) defined as $M_{i,j} = (j - i)^2$ and observe that $M$ is the element-wise square of a matrix $H_{i,j} = j - i$. We have $H = P_N Q_N$, where

$$P_N = \begin{pmatrix} 1 & 0 \\ 1 & -1 \\ 1 & -2 \\ \vdots & \vdots \\ 1 & -2^N + 1 \end{pmatrix}, Q_N = \begin{pmatrix} 0 & 1 & 2 & \cdots & 2^N - 1 \\ 1 & 1 & 1 & \cdots & 1 \end{pmatrix}. \tag{S63}$$

Now consider the tensors

$$A_{1,0} = \begin{pmatrix} 1 & 0 \end{pmatrix}, \;\; A_{1,1} = \begin{pmatrix} 1 & -1 \end{pmatrix}, \tag{S64}$$

$$\forall n > 1, \;\; A_{n,0} = \begin{pmatrix} 1 & 0 \\ 0 & 1 \end{pmatrix}, \;\; A_{n,1} = \begin{pmatrix} 1 & -2^n \\ 0 & 1 \end{pmatrix}, \tag{S65}$$

and build an MPS by contracting tensors $A_1$ to $A_n$. We obtain a tensor $T_N$ with $N$ open indices corresponding to $N$ binary variables and an extra virtual index of dimension 2, as in Equation (S61). If we reshape this tensor as a $2^N \times 2$ matrix we obtain matrix $P_N$. In the same way we can obtain an MPS of non-negative tensors such that contracting $N$ sites gives a tensor that can be reshaped as $Q_N$. Contracting the two extra virtual indices between the two MPS, we finally obtain an $\text{MPS}_{\mathbb{R}}$ over $2N$ binary variables such that $H$ is the central bipartition of the resulting tensor. By squaring this MPS we obtain a $\text{BM}_{\mathbb{R}}$ over $2N$ binary variables such that $M$ is the central bipartition of the resulting tensor. Suppose that there is an $\text{MPS}_{\mathbb{R}_{\geq 0}}$ defining the same probability mass function over $2N$ variables, then it has TT-rank$_{\mathbb{R}_{\geq 0}}$ larger or equal to the non-negative rank of $M$, which is larger than $\log_2 2^N = N$. This proves the case $[2, 3]$, and again the remaining cases follow from Proposition 1. $\qquad\square$

**Proposition 7** ($[3, 4]$)**.** *There exists a family of non-negative tensors over $2N$ binary variables with constant Born-rank$_{\mathbb{C}} = 2$, but with Born-rank$_{\mathbb{R}} \geq N$.*

*Proof.* Consider the $N \times N$ matrices $M_N$ from Lemma 15. These matrices have complex square root rank 2 but real square root rank $N$. Using Lemma 16, this means that there is a $\text{BM}_{\mathbb{C}}$ over $2N$ variables of Born-rank$_{\mathbb{C}}$ equal to 2 such that $M_N$ is a submatrix of the central bipartition of the resulting tensor. The Born-rank$_{\mathbb{R}}$ of this tensor is at least the Born-rank$_{\mathbb{R}}$ of $M_N$, which is $N$. $\qquad\square$

## 3 Learning algorithms and numerical experiments

### 3.1 Learning algorithms for LPS

Consider first the setting in which one is given samples $\{\mathbf{x_i} = (X_1^i, \ldots, X_N^i)\}$ from a discrete multivariate distribution and would like to obtain an efficient approximation of this distribution. This can be done by minimizing the negative log-likelihood,

$$L = -\sum_i \log \frac{T_{\mathbf{x_i}}}{Z_T}, \tag{S66}$$

where $i$ indexes training samples and $T_{\mathbf{x_i}}$ is given by the contraction of one of the tensor-network models we have introduced. The derivative of the log-likelihood with respect to a parameter $w$ in the tensor network is given by

$$\partial_w L = -\sum_i \frac{\partial_w T_{\mathbf{x_i}}}{T_{\mathbf{x_i}}} - \frac{\partial_w Z_T}{Z_T}. \tag{S67}$$

The negative log-likelihood can be minimized using a mini-batch gradient-descent algorithm. At each step of the optimization, the sum is computed over a batch of training instances. The parameters in the tensor network are then updated by a small step in the inverse direction of the gradient. Note that when using complex tensors, the derivatives are replaced by Wirtinger derivatives with respect to the conjugated tensor elements. This algorithm requires the computation of $T_{\mathbf{x_i}}$ and $\partial_w T_{\mathbf{x_i}}$ for a training instance, as well as of $Z_T$ and $\partial_w Z_T$.

We first focus on the computation of these quantities for LPS. Since Born machines are LPS of purification dimension $\mu = 1$, they can directly use the same algorithm [11]. For an LPS$_\mathbb{C}$ of puri-rank $r$, the normalization $Z_T$ can be computed by contracting the tensor network

$$Z_T = \sum_{X_1,\ldots,X_N} T_{X_1,\ldots,X_N} = \ \ \text{(S68)}$$

This contraction is performed in $\mathcal{O}(d\mu r^3 N)$ operations from left to right by contracting at each step the two vertical indices and then each of the two horizontal indices. During this contraction, intermediate results from the contraction of the first $i$ tensors are stored in $E_i$, and the same procedure is repeated from the right with intermediate results of the contraction of the last $N - i$ tensors stored in $F_{i+1}$. The derivatives of the normalization for each tensor are then computed as

$$\frac{\partial Z_T}{\partial \bar{A}_{i,m}^{j,k,l}} = \ \ \text{(S69)}$$

which also costs $\mathcal{O}(d\mu r^3 N)$ operations. Computing $T_{\mathbf{x_i}}$ for a training example and its derivative is done in the same way, except that the contracted index corresponding to an observed variable is now fixed to its observed value.

Note that here the training is done by computing the gradients of the log-likelihood over all tensors for each batch of training example and then updating all tensors at once in a gradient-descent optimization scheme. A different approach would be a DMRG-like algorithm where only a few tensors are updated at a time. The computation of $Z_T$ and its derivative may be greatly simplified by using canonical forms [4].

## 3.2 Learning algorithms for MPS$_{\mathbb{R}_{\geq 0}}$

As in the case of LPS, MPS$_{\mathbb{R}_{\geq 0}}$ can be trained using gradient descent to minimize the log-likelihood. Consider an MPS$_{\mathbb{R}_{\geq 0}}$, and assume that the tensors $A_i$ in the MPS are given by the element-wise square of real tensors $B_i$. The normalization can be computed by contracting the following tensor network from left to right, where the circles represent a vector of ones of dimension $d$, as

$$Z_T = \ \ \text{(S70)}$$

This contraction is performed in $\mathcal{O}(dr^2 N)$ operations. During this contraction, intermediate results from the contraction of the first $i$ tensors are stored in $E_i$, and the same procedure is repeated from the right with intermediate results of the contraction of the last $N - i$ tensors stored in $F_{i+1}$. The derivatives of the normalization for each tensor are then computed as

$$\frac{\partial Z_T}{\partial A_{i,m}^{j,k}} = \ \ \text{(S71)}$$

and the derivatives with respect to the original parameters are obtained as

$$\frac{\partial Z_T}{\partial B_{i,m}^{j,k}} = 2 \frac{\partial Z_T}{\partial A_{i,m}^{j,k}} B_{i,m}^{j,k}. \qquad \text{(S72)}$$

Applying the same procedure by replacing the circle tensors with indices corresponding to the observed variables at a training example leads to the computation of $T_{\mathbf{x_i}}$ and its derivative.

## 3.3 Tensor factorizations

We minimize $D(P||T/Z_T)$ through a non-linear limited-memory BFGS optimization algorithm. The gradient of the KL-divergence depends on the log-derivatives of $T$ and of $Z_T$, which have already been obtained while computing the gradient of the log-likelihood.

Optimizing an $MPS_\mathbb{R}$ requires to impose the non-negativity of the tensor network. In order to still provide a comparison with $MPS_\mathbb{R}$, we optimize them by adding a penalty term constraining all elements of the contracted tensor to be non-negative. This constraint is difficult to satisfy, so the optimization may not converge to the global minimum.

Note that we could have chosen any distance instead of the KL-divergence. In particular if instead we minimize the 2-norm for vectors $||P - T||_2$ the optimization for an $MPS_\mathbb{R}$ can be done by keeping only the largest singular values of the tensor $P$ (in the case of matrices), and a good starting point for tensors can be obtained through successive truncated singular value decompositions. We find that the results we obtained are not significantly modified if the 2-norm was considered instead of the KL-divergence.

## 3.4 Generalization performance

Our results focus on the expressive power of different tensor-network representations, and for this reason the numerical experiments focused on the accuracy obtained while training on the full datasets. It is nevertheless also interesting to see how much these results would be modified if we considered generalization performance instead.

In order to investigate the differences obtained on training sets or test sets, we focus on the biofam data set of family life states from the Swiss Household Panel biographical survey [12], since small bond dimensions are sufficient to get already converged results on training sets and since this data set includes variables that have a natural sequential order : they represent the family life states from age 15 to 30 (sequence length is 16) and the variables can take 8 possible values. The probability distribution is therefore a tensor of size $8^{16} = 2^{48}$, which requires to use models with smaller number of parameters. We use 1000 examples in the training set, 500 in the validation set and 500 in the test set. All models are trained for 20000 epochs with a batch size of 20 on the training set, and the best training accuracy with respect to different learning rates (using a grid search on powers of 10 going from $10^{-5}$ to $10^5$) is recorded. The procedure is repeated 20 times with a different random initialization of the tensors, and the mean and standard deviation are reported in Fig. S2 (left). For each initial condition, the best model on the validation set is chosen to be evaluated on the test set. The mean and standard deviation of the accuracy on the test set with respect to different initial conditions are reported Fig. S2 (right).

Figure S2: Negative log-likelihood per sample obtained on the training set (left) and on the test set (right) for different tensor-network representations on the biofam data set. The error bars represent one standard deviation with different initial conditions.

The results indicate that despite rather small differences between HMM and non-negative MPS on the training set, the differences on the test set are important and non-negative MPS are able to reach a better accuracy at larger ranks, when HMM do not improve anymore. Born machines and LPS all reach better performance than HMM and non-negative MPS already for small ranks, and with smaller variance with respect to the initial conditions.

## Footnotes

\*Corresponding author, `ivan.glasser@mpq.mpg.de`