[Reviews · NeurIPS 2019]

Reviewer 1



The authors compare the ranks of tensor representations of HMM, and outputs of quantum circuits with two qubit unitary gates yielding Matrix product States (MPS) and so-called Locally Purified States (LPS) when ancillary unmeasured bits are present. A general comment: Born machines automaticaly enforce positivity but is it clear that 83) and (4) are less than 1 ? And sum to 1 ? The A's come from some unitary circuits in SM ? If yes the main problem formulation seems not selfcontained in sect.2. Explanations or rewiting welcome ! -- Some of the results seems a bit obvious or intuitive. Some are more surprizing namely the very large (at least of the order of the number of qubits) difference in rank when one works in the real field versus complex field. -- That there is a gap of expressivity between the two fields is not surprizing (maybe I am naive here) but the largeness of the gap is puzzling. Can you comment on your intuitions ? Is this due to very specific examples constructed in the SM ? Or is it a more generic feature ? These comments would also improve clarity. -- In the same vein that TT rank is smaller than purirank etc (proposition 4) was counterintuitive to me at first. As said later the catch is probably that it is difficult to enforce the positivity constraint. But such intuitions could be given much earlier after the statements of each proposition. -- Further results about the growth rate of this gap as a function of N would be interesting, but these are not provided. I understand this is left as future work. Reasonable and not a problem. -- I did not follow all proofs in detail but the parts I read seem sound. -- To improve the clarity of the presentation a better explanation of the link between figure 3 and Proposition 1 would be welcome. A few sentence would suffice to develop a better intuition. Review update: the authors have responded to most of these questions and propose added explanations in the final version. In my opinion they have also addressed concerns of other reviewers. The responses are convincing and I trust the final version of the paper will improve from the pedagogic stance.

Reviewer 2



==== Summary ==== The paper builds on the known relations between tensor factorizations and probabilistic models, and analyze the expressivity of two families of probabilistic models over sequences. The paper focuses on tensor networks (TN), a graphical language for describing most known forms of tensor factorizations, and specifically, variants or extensions of what is known as a Matrix Product State (or Tensor Train) factorization, which was previously shown to be a generalization of Hidden Markov Models. The paper examines two known methods of leveraging these kinds of factorizations for probabilistic modeling of sequences (ensuring they represent a non-negative tensor): (a) limiting the factors to be non-negative, which includes as a special case HMMs, (b) Born Machines, which represent the probability as the squared magnitude of the factorized tensor (whose factors can be real or complex), where the element-wise squared magnitude tensor can be represented natively using a different TN induced by the original MPS. In addition to these two methods, the authors also suggest the use of another TN called Locally Purified State (LPS), which can be seen as a generalization of the Born Machine induced TN. The authors characterize the expressivity and compare these three methods (NNTF, BM, and LPS) through relations to different kinds of algebraic rank measures. Similar analysis methods were previously used to analyze the expressiveness of deep learning methods [1,2], as the authors explain. Their main result is showing that neither NTFF or BM are generally preferred because for each, there are special cases where they are significantly more expressive than the other. This non-preferability is the basis on which the authors propose the use of LPS instead, which is shown to be theoretically better than the other two methods. The LPS TN was previously only used in the context of studying quantum systems. Finally, the authors validate their theoretical results by demonstrating on synthetic data and basic datasets that LPS has better expressiveness characteristics than NNTF (including HMMs) and BM. ==== Detailed Review ==== This is a good theoretical paper, and though it only has modest, practical implications, I believe it has a place in NeurIPS. While the connections between tensor factorization and probabilistic models have been known for quite some time, there are relatively few papers which leverage this connection to gain theoretical insight into these models. Previous works have focused on specific non-negative tensor factorizations, e.g., [3] studies non-negative Hierarchical Tucker and CP decompositions, and [4] extends it to TT / MPS, and includes approximation analysis, but none have studied the use of Born Machines or how they compare to NNTF. Though this paper uses MPS and sequential probabilistic modeling as a running example, it seems clear that many of its points could be generalized to other factorization schemes, and so its main result on comparing NNTF vs. Born Machines could be taken as a general comparison of these two approaches, and not just specific to MPS (as the authors mention as an aside). The main results show that no one approach is better than the other and that they both suffer from different limitations, is clearly of significance, and could spark further development on how to overcome this gap. The method proposed by authors seem to overcome these theoretical limitations; however, it is unclear from the experiments if it manifests in actual performance benefits on large scale datasets. Nevertheless, the basic experiments presented in the paper are sufficient for merely validating that the theoretical findings could lead to practical benefits. Despite the above, there were a few issues that I did find with the paper: 1) The paper also shows that using complex values in BM leads to expressivity gains compared to real BM, and then extrapolates to the general case of complex-valued Neural Networks and other ML models. While The proof for BMs is correct, it is misleading to attribute the expressivity gains to the use of complex values. The source of the additional expressivity is actually due to the definition of the magnitude of complex numbers as the norm of the real and imaginary components, and not to the complex multiplication operations of the factorization. Simply using real BMs outputting vectors instead of scalars and taking their norm would result in the same benefits (because complex matrix operations can be mapped to real matrix operations), and could actually be generalized to dim > 2, and so lead to a further boost to expressiveness. I would strongly recommend the authors to either explicitly mention the limitations of their results for extrapolation to general cases, or remove this as a possible implication of their results. 2) The paper would benefit from a short discussion on how tensor factorizations and probabilistic models are also deeply related to Sum-Product Networks and Arithmetic Circuits. This should probably be mentioned as related works, and specifically, cite [3,4]. Moreover, the separation between TT_R and TT_R>0 can actually be easily derived from a similar claim on the separation between general Arithmetic Circuits, and Monotone Arithmetic Circuits (AC with positive weights), see [5] for details. 3) Though the current experiments are sufficient, they are just the bare minimum for showing the theory could be relevant to practitioners. However, it would significantly improve the paper also to add comparisons against other competing methods, even ones which are not directly comparable or even favorable (e.g., general SPNs and GMMs which are also tractable, and maybe even autoregressive models). [1] Cohen et al. Analysis and design of convolutional networks via hierarchical tensor decompositions. [2] Khrulkov et al. Expressive power of recurrent neural networks. [3] Sharir et al. Tensorial Mixture Models. [4] Jaini et al. Deep Homogeneous Mixture Models: Representation, Separation, and Approximation. [5] Shpilka et al. Arithmetic Circuits: a survey of recent results and open questions. === Update following rebuttal === The authors have acknowledge the issues raised in my review, and plan to add clarifications to the paper to address them. I stand by my original review that this paper provides an interesting and novel analysis of certain probabilistic models, and recommend accepting it. I believe my original score reflects this assessment, and so remains unchanged.

Reviewer 3



This paper is original and analyzed several tensor network's expressive power. The paper presents MPS, BM, and LPS to model multivariate probability mass function. For MPS, it is well-known that can be used to represent HMM or other probabilistic models, while BM and LPS are relatively new but straightforward. The learning algorithms are based on minimization of negative log-likelihood, which is very natural and straightforward. In the paper, 7 proposition are given, which are the main contributions and results. These propositions mostly focus on analyzing the rank relations among several tensor networks. These results are important and interesting. Weakness: 1. Although these tensor networks can be used to represent PMF of discrete variables. How these results are useful to machine learning algorithms or analyze the algorithm is not clear. Hence, the significance of this paper is poor. 2. The two experiments are all based on very small dataset either generated or realistic data. The evaluation is performed on KL-divergence or NLL, which only show how good the model can fit the data, rather than generalization performance. How these results are useful for machine learning? In addition, MPS, BM, LPS are quite similar in the structure. There are many well known tensor models, CP, Tucker, Hirachical Tucker are not compared. There are also more complicated models like MERA, PEPS. I have read the authors' rebuttal, they addressed some of questions well. But the generalization is not considered, thus it becomes a standard non-negative tensor factorization problem on the PMF data. Hence, I will remain the original score.

[Author Response · NeurIPS 2019]

We would like to thank all reviewers for their insightful comments and suggestions, and provide replies below.

**Reviewer 1**

**Settings and normalization.** The tensors $T$ in equations (2)-(4) may not be normalized. However, in this work we always consider models for probability mass functions of the form $T/Z_T$, where $Z_T$ is the normalization factor (which can be computed efficiently). Additionally, the $A$'s in the definitions are arbitrary tensors containing the free parameters of the model. They do not have to originate from quantum circuits, but for any quantum circuit one can define $A$'s such that it is equivalent to a BM. To enhance the clarity of the manuscript, we will add a paragraph at the beginning of Section 2 expanding on the above explanations and detailing the settings and requirements on the tensor networks.

**Link between Figure 3 and Proposition 1.** We will add the following explanation of the relationship between the expressive power, the ranks, Figure 3 and Proposition 1: *For a given rank, there is a set of non-negative tensors that can be exactly represented by a given tensor network, and as the rank increases, this set grows. These sets are represented in Fig. 3 for the case in which the ranks of the different tensor networks are equal. When one set is included in another, it means that for every non-negative tensor, the rank of one of the tensor-network factorizations is always greater than or equal to the rank of the other factorization. The inclusion relationships between these sets can therefore be characterized in terms of inequalities between the ranks, as detailed in Proposition 1.*

**Intuition behind propositions.** Given the above explanation, Proposition 3 can be intuitively understood to ask by how much one needs to increase the rank of a tensor network such that the set of non-negative tensors it can represent includes the set corresponding to another tensor network. We will expand on this in the text and include intuition for the remaining propositions. In particular, the separations between $\text{MPS}_{\mathbb{R}\geq 0}$ and BM arise from the difference in ranks between probability distributions and square-roots of probability distributions, and the separation between real or complex BM comes from the combination of real and imaginary parts through the modulus squared (see also the reply to Reviewer 2). The growth rate of these separations is lower-bounded in Propositions 4-7. An upper bound is not available, as there could be other distributions providing larger separations than the ones we have found.

**Reviewer 2**

**On the role of complex numbers.** We agree that the separation between real and complex BM comes from the use of the modulus of complex numbers in these specific networks. As pointed out, "using real BMs outputting vectors [...] would result in the same benefits": This is correct and precisely why a real LPS of purification dimension 2 includes a complex BM. This fact (shown in Table 2) will be highlighted in the text to provide a correct interpretation for this result. We acknowledge that an expressivity advantage due to complex numbers cannot be extrapolated to general cases such as neural networks. This limitation will be included in the paper.

**Relationship with Sum-Product Networks and Arithmetic Circuits.** We would like to thank the reviewer for providing these references and we will include a paragraph on these relationships and previously obtained results.

**Numerical experiments.** We agree that the numerics do not demonstrate the practical advantage of LPS in real-world problems, but rather provide evidence that the theoretical results hold also for distributions that have not been fine-tuned. We aim to investigate their performance on real-world datasets in future work, which might require further research, for example on the use of these tensor networks with continuous variables. In order to provide a comparison, we will add an indication on Fig. 5 of the accuracy of the optimal Bayesian network without hidden variables, where the network graph is learned from the data. This includes simple autoregressive models and avoids hyper-parameter and architecture tuning. It reaches a negative log-likelihood of $5.8$, $13.4$, $10.4$, $9.9$, $8.7$ and $6.0$ on datasets (a)-(f) respectively.

**Reviewer 3**

**Generalization performance.** We agree that generalization performance is a very important topic, and that relating generalization, either heuristically or analytically, to quantities such as the rank of these models would be highly desirable. As our analysis is focused on expressive power, evaluation of the models on training sets is useful for validating our theoretical results in practical settings. In order to investigate generalization performance we will add a plot in the supplementary material showing the test set accuracy of these models with respect to the rank. On the biofam dataset the lowest negative log-likelihood on the test set attained by an LPS is $6.4$, while for an HMM it is $7.4$.

**Usefulness for machine learning algorithms.** The tensor networks we consider are a class of probabilistic models which admit efficient learning, inference and sampling algorithms, and can therefore be used for the same ML tasks as HMMs while having some expressivity advantages. Indeed, it remains unclear whether this method can lead to state-of-the-art performances, but our theoretical results show that this is worth investigating in the future. Non-negative tensor factorizations are also used in diverse areas of ML such as recommendation systems or signal processing, and the factorizations we introduce may be useful in this context. Moreover, our results and techniques can be straightforwardly generalized to other tensor networks and interpreted as a general comparison between different strategies for ensuring non-negativity of a tensor factorization. We will add a paragraph expanding upon this in the paper.

[Meta-Review · NeurIPS 2019]

This paper compares the expressive power of different classes of tensor networks for factorizing high-dimensional discrete probability distributions. All reviewers agree that this is a solid theoretical contribution that is of interest to the community.